# INDUCTIVE BIAS OF DEEP CONVOLUTIONAL NETWORKS THROUGH POOLING GEOMETRY

**Nadav Cohen & Amnon Shashua**
`{cohennadav,shashua}@cs.huji.ac.il`

## ABSTRACT

Our formal understanding of the inductive bias that drives the success of convolutional networks on computer vision tasks is limited. In particular, it is unclear what makes hypotheses spaces born from convolution and pooling operations so suitable for natural images. In this paper we study the ability of convolutional networks to model correlations among regions of their input. We theoretically analyze convolutional arithmetic circuits, and empirically validate our findings on other types of convolutional networks as well. Correlations are formalized through the notion of separation rank, which for a given partition of the input, measures how far a function is from being separable. We show that a polynomially sized deep network supports exponentially high separation ranks for certain input partitions, while being limited to polynomial separation ranks for others. The network's pooling geometry effectively determines which input partitions are favored, thus serves as a means for controlling the inductive bias. Contiguous pooling windows as commonly employed in practice favor interleaved partitions over coarse ones, orienting the inductive bias towards the statistics of natural images. Other pooling schemes lead to different preferences, and this allows tailoring the network to data that departs from the usual domain of natural imagery. In addition to analyzing deep networks, we show that shallow ones support only linear separation ranks, and by this gain insight into the benefit of functions brought forth by depth – they are able to efficiently model strong correlation under favored partitions of the input.

## 1 INTRODUCTION

A central factor in the application of machine learning to a given task is the *inductive bias*, *i.e.* the choice of hypotheses space from which learned functions are taken. The restriction posed by the inductive bias is necessary for practical learning, and reflects prior knowledge regarding the task at hand. Perhaps the most successful exemplar of inductive bias to date manifests itself in the use of convolutional networks (LeCun and Bengio (1995)) for computer vision tasks. These hypotheses spaces are delivering unprecedented visual recognition results (*e.g.* Krizhevsky et al. (2012); Szegedy et al. (2015); Simonyan and Zisserman (2014); He et al. (2015)), largely responsible for the resurgence of deep learning (LeCun et al. (2015)). Unfortunately, our formal understanding of the inductive bias behind convolutional networks is limited – the assumptions encoded into these models, which seem to form an excellent prior knowledge for imagery data, are for the most part a mystery.

Existing works studying the inductive bias of deep networks (not necessarily convolutional) do so in the context of *depth efficiency*, essentially arguing that for a given amount of resources, more layers result in higher expressiveness. More precisely, depth efficiency refers to a situation where a function realized by a deep network of polynomial size, requires super-polynomial size in order to be realized (or approximated) by a shallower network. In recent years, a large body of research was devoted to proving existence of depth efficiency under different types of architectures (see for example Delalleau and Bengio (2011); Pascanu et al. (2013); Montufar et al. (2014); Telgarsky (2015); Eldan and Shamir (2015); Poggio et al. (2015); Mhaskar et al. (2016)). Nonetheless, despite the wide attention it is receiving, depth efficiency does not convey the complete story behind the inductive bias of deep networks. While it does suggest that depth brings forth functions that are otherwise unattainable, it does not explain why these functions are useful. Loosely speaking, the

hypotheses space of a polynomially sized deep network covers a small fraction of the space of all functions. We would like to understand why this small fraction is so successful in practice.

A specific family of convolutional networks gaining increased attention is that of *convolutional arithmetic circuits*. These models follow the standard paradigm of locality, weight sharing and pooling, yet differ from the most conventional convolutional networks in that their point-wise activations are linear, with non-linearity originating from product pooling. Recently, Cohen et al. (2016b) analyzed the depth efficiency of convolutional arithmetic circuits, showing that besides a negligible (zero measure) set, all functions realizable by a deep network require exponential size in order to be realized (or approximated) by a shallow one. This result, termed *complete depth efficiency*, stands in contrast to previous depth efficiency results, which merely showed *existence* of functions efficiently realizable by deep networks but not by shallow ones. Besides their analytic advantage, convolutional arithmetic circuits are also showing promising empirical performance. In particular, they are equivalent to SimNets – a deep learning architecture that excels in computationally constrained settings (Cohen and Shashua (2014); Cohen et al. (2016a)), and in addition, have recently been utilized for classification with missing data (Sharir et al. (2016)). Motivated by these theoretical and practical merits, we focus our analysis in this paper on convolutional arithmetic circuits, viewing them as representative of the class of convolutional networks. We empirically validate our conclusions with both convolutional arithmetic circuits and *convolutional rectifier networks* – convolutional networks with rectified linear (ReLU, Nair and Hinton (2010)) activation and max or average pooling. Adaptation of the formal analysis to networks of the latter type, similarly to the adaptation of the analysis in Cohen et al. (2016b) carried out by Cohen and Shashua (2016), is left for future work.

Our analysis approaches the study of inductive bias from the direction of function inputs. Specifically, we study the ability of convolutional arithmetic circuits to model correlation between regions of their input. To analyze the correlations of a function, we consider different partitions of input regions into disjoint sets, and ask how far the function is from being separable w.r.t. these partitions. Distance from separability is measured through the notion of *separation rank* (Beylkin and Mohlenkamp (2002)), which can be viewed as a surrogate of the $L^2$ distance from the closest separable function. For a given function and partition of its input, high separation rank implies that the function induces strong correlation between sides of the partition, and vice versa.

We show that a deep network supports exponentially high separation ranks for certain input partitions, while being limited to polynomial or linear (in network size) separation ranks for others. The network's pooling geometry effectively determines which input partitions are favored in terms of separation rank, *i.e.* which partitions enjoy the possibility of exponentially high separation rank with polynomial network size, and which require network to be exponentially large. The standard choice of square contiguous pooling windows favors interleaved (entangled) partitions over coarse ones that divide the input into large distinct areas. Other choices lead to different preferences, for example pooling windows that join together nodes with their spatial reflections lead to favoring partitions that split the input symmetrically. We conclude that in terms of modeled correlations, pooling geometry controls the inductive bias, and the particular design commonly employed in practice orients it towards the statistics of natural images (nearby pixels more correlated than ones that are far apart). Moreover, when processing data that departs from the usual domain of natural imagery, prior knowledge regarding its statistics can be used to derive respective pooling schemes, and accordingly tailor the inductive bias.

With regards to depth efficiency, we show that separation ranks under favored input partitions are exponentially high for all but a negligible set of the functions realizable by a deep network. Shallow networks on the other hand, treat all partitions equally, and support only linear (in network size) separation ranks. Therefore, almost all functions that may be realized by a deep network require a replicating shallow network to have exponential size. By this we return to the complete depth efficiency result of Cohen et al. (2016b), but with an added important insight into the benefit of functions brought forth by depth – they are able to efficiently model strong correlation under favored partitions of the input.

The remainder of the paper is organized as follows. Sec. 2 provides a brief presentation of necessary background material from the field of tensor analysis. Sec. 3 describes the convolutional arithmetic circuits we analyze, and their relation to tensor decompositions. In sec. 4 we convey the concept of separation rank, on which we base our analyses in sec. 5 and 6. The conclusions from our analyses are empirically validated in sec. 7. Finally, sec. 8 concludes.

## 2 PRELIMINARIES

The analyses carried out in this paper rely on concepts and results from the field of tensor analysis. In this section we establish the minimal background required in order to follow our arguments [1] , referring the interested reader to Hackbusch (2012) for a broad and comprehensive introduction to the field.

The core concept in tensor analysis is a *tensor*, which for our purposes may simply be thought of as a multi-dimensional array. The *order* of a tensor is defined to be the number of indexing entries in the array, which are referred to as *modes*. The *dimension* of a tensor in a particular mode is defined as the number of values that may be taken by the index in that mode. For example, a 4-by-3 matrix is a tensor of order 2, *i.e.* it has two modes, with dimension 4 in mode 1 and dimension 3 in mode 2. If $\mathcal{A}$ is a tensor of order $N$ and dimension $M_i$ in each mode $i \in [N] := \{1, \ldots, N\}$, the space of all configurations it can take is denoted, quite naturally, by $\mathbb{R}^{M_1 \times \cdots \times M_N}$.

A fundamental operator in tensor analysis is the *tensor product*, which we denote by $\otimes$. It is an operator that intakes two tensors $\mathcal{A} \in \mathbb{R}^{M_1 \times \cdots \times M_P}$ and $\mathcal{B} \in \mathbb{R}^{M_{P+1} \times \cdots \times M_{P+Q}}$ (orders $P$ and $Q$ respectively), and returns a tensor $\mathcal{A} \otimes \mathcal{B} \in \mathbb{R}^{M_1 \times \cdots \times M_{P+Q}}$ (order $P + Q$) defined by: $(\mathcal{A} \otimes \mathcal{B})_{d_1 \ldots d_{P+Q}} = \mathcal{A}_{d_1 \ldots d_P} \cdot \mathcal{B}_{d_{P+1} \ldots d_{P+Q}}$. Notice that in the case $P = Q = 1$, the tensor product reduces to the standard outer product between vectors, *i.e.* if $\mathbf{u} \in \mathbb{R}^{M_1}$ and $\mathbf{v} \in \mathbb{R}^{M_2}$, then $\mathbf{u} \otimes \mathbf{v}$ is no other than the rank-1 matrix $\mathbf{u}\mathbf{v}^\top \in \mathbb{R}^{M_1 \times M_2}$.

We now introduce the important concept of matricization, which is essentially the rearrangement of a tensor as a matrix. Suppose $\mathcal{A}$ is a tensor of order $N$ and dimension $M_i$ in each mode $i \in [N]$, and let $(I, J)$ be a partition of $[N]$, *i.e.* $I$ and $J$ are disjoint subsets of $[N]$ whose union gives $[N]$. We may write $I = \{i_1, \ldots, i_{|I|}\}$ where $i_1 < \cdots < i_{|I|}$, and similarly $J = \{j_1, \ldots, j_{|J|}\}$ where $j_1 < \cdots < j_{|J|}$. The *matricization of $\mathcal{A}$ w.r.t. the partition* $(I, J)$, denoted $[\![\mathcal{A}]\!]_{I,J}$, is the $\prod_{t=1}^{|I|} M_{i_t}$-by-$\prod_{t=1}^{|J|} M_{j_t}$ matrix holding the entries of $\mathcal{A}$ such that $\mathcal{A}_{d_1 \ldots d_N}$ is placed in row index $1 + \sum_{t=1}^{|I|}(d_{i_t} - 1) \prod_{t'=t+1}^{|I|} M_{i_{t'}}$ and column index $1 + \sum_{t=1}^{|J|}(d_{j_t} - 1) \prod_{t'=t+1}^{|J|} M_{j_{t'}}$. If $I = \emptyset$ or $J = \emptyset$, then by definition $[\![\mathcal{A}]\!]_{I,J}$ is a row or column (respectively) vector of dimension $\prod_{t=1}^{N} M_t$ holding $\mathcal{A}_{d_1 \ldots d_N}$ in entry $1 + \sum_{t=1}^{N}(d_t - 1) \prod_{t'=t+1}^{N} M_{t'}$.

A well known matrix operator is the *Kronecker product*, which we denote by $\odot$. For two matrices $A \in \mathbb{R}^{M_1 \times M_2}$ and $B \in \mathbb{R}^{N_1 \times N_2}$, $A \odot B$ is the matrix in $\mathbb{R}^{M_1 N_1 \times M_2 N_2}$ holding $A_{ij} B_{kl}$ in row index $(i - 1)N_1 + k$ and column index $(j - 1)N_2 + l$. Let $\mathcal{A}$ and $\mathcal{B}$ be tensors of orders $P$ and $Q$ respectively, and let $(I, J)$ be a partition of $[P + Q]$. The basic relation that binds together the tensor product, the matricization operator, and the Kronecker product, is:

$$[\![\mathcal{A} \otimes \mathcal{B}]\!]_{I,J} = [\![\mathcal{A}]\!]_{I \cap [P], J \cap [P]} \odot [\![\mathcal{B}]\!]_{(I-P) \cap [Q], (J-P) \cap [Q]} \qquad (1)$$

where $I - P$ and $J - P$ are simply the sets obtained by subtracting $P$ from each of the elements in $I$ and $J$ respectively. In words, eq. 1 implies that the matricization of the tensor product between $\mathcal{A}$ and $\mathcal{B}$ w.r.t. the partition $(I, J)$ of $[P + Q]$, is equal to the Kronecker product between two matricizations: that of $\mathcal{A}$ w.r.t. the partition of $[P]$ induced by the lower values of $(I, J)$, and that of $\mathcal{B}$ w.r.t. the partition of $[Q]$ induced by the higher values of $(I, J)$.

## 3 CONVOLUTIONAL ARITHMETIC CIRCUITS

The convolutional arithmetic circuit architecture on which we focus in this paper is the one considered in Cohen et al. (2016b), portrayed in fig. 1(a). Instances processed by a network are represented as $N$-tuples of $s$-dimensional vectors. They are generally thought of as images, with the $s$-dimensional vectors corresponding to local patches. For example, instances could be 32-by-32 RGB images, with local patches being $5 \times 5$ regions crossing the three color bands. In this case, assuming a patch is taken around every pixel in an image (boundaries padded), we have $N = 1024$ and $s = 75$. Throughout the paper, we denote a general instance by $X = (\mathbf{x}_1, \ldots, \mathbf{x}_N)$, with $\mathbf{x}_1 \ldots \mathbf{x}_N \in \mathbb{R}^s$ standing for its patches.

---

[1] The definitions we give are actually concrete special cases of more abstract algebraic definitions as given in Hackbusch (2012). We limit the discussion to these special cases since they suffice for our needs and are easier to grasp.

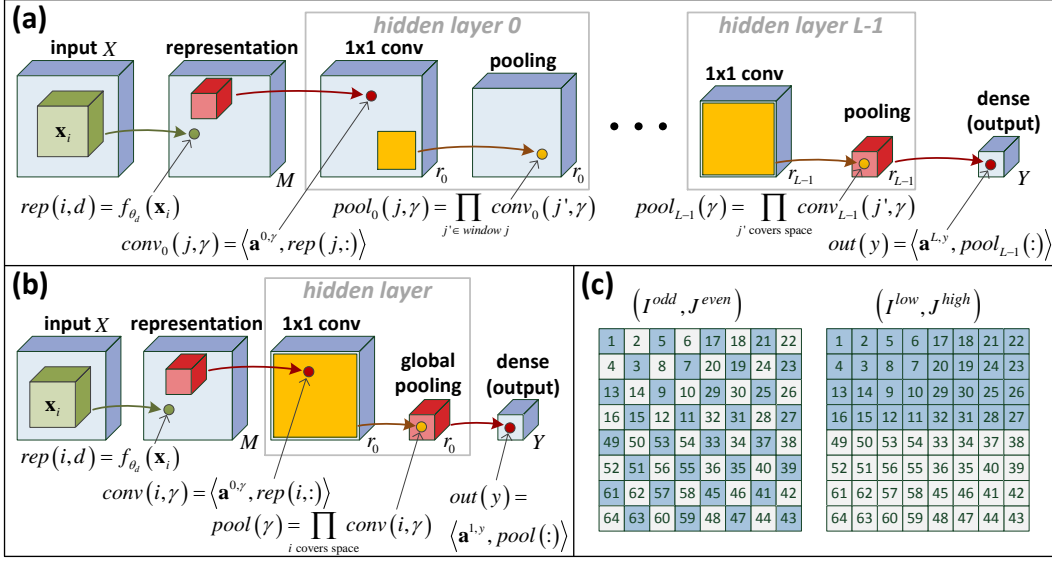

Figure 1: Best viewed in color. *(a)* Convolutional arithmetic circuit architecture analyzed in this paper (see description in sec. 3). *(b)* Shallow network with global pooling in its single hidden layer. *(c)* Illustration of input patch ordering for deep network with $2 \times 2$ pooling windows, along with patterns induced by the partitions $(I^{odd}, J^{even})$ and $(I^{low}, J^{high})$ (eq. 8 and 9 respectively).

The first layer in a network is referred to as *representation*. It consists of applying $M$ *representation functions* $f_{\theta_1} \dots f_{\theta_M} : \mathbb{R}^s \to \mathbb{R}$ to all patches, thereby creating $M$ feature maps. In the case where representation functions are chosen as $f_{\theta_d}(\mathbf{x}) = \sigma(\mathbf{w}_d^\top \mathbf{x} + b_d)$, with parameters $\theta_d = (\mathbf{w}_d, b_d) \in \mathbb{R}^s \times \mathbb{R}$ and some point-wise activation $\sigma(\cdot)$, the representation layer reduces to a standard convolutional layer. More elaborate settings are also possible, for example modeling the representation as a cascade of convolutional layers with pooling in-between. Following the representation, a network includes $L$ hidden layers indexed by $l = 0 \dots L-1$. Each hidden layer $l$ begins with a $1 \times 1$ *conv* operator, which is simply a three-dimensional convolution with $r_l$ channels and filters of spatial dimensions 1-by-1. [2] This is followed by spatial pooling, that decimates feature maps by taking products of non-overlapping two-dimensional windows that cover the spatial extent. The last of the $L$ hidden layers ($l = L-1$) reduces feature maps to singletons (its pooling operator is global), creating a vector of dimension $r_{L-1}$. This vector is mapped into $Y$ network outputs through a final dense linear layer.

Altogether, the architectural parameters of a network are the type of representation functions ($f_{\theta_d}$), the pooling window shapes and sizes (which in turn determine the number of hidden layers $L$), and the number of channels in each layer ($M$ for representation, $r_0 \dots r_{L-1}$ for hidden layers, $Y$ for output). Given these architectural parameters, the learnable parameters of a network are the representation weights ($\theta_d$ for channel $d$), the conv weights ($\mathbf{a}^{l,\gamma}$ for channel $\gamma$ of hidden layer $l$), and the output weights ($\mathbf{a}^{L,y}$ for output node $y$).

For a particular setting of weights, every node (neuron) in a given network realizes a function from $(\mathbb{R}^s)^N$ to $\mathbb{R}$. The *receptive field* of a node refers to the indexes of input patches on which its function may depend. For example, the receptive field of node $j$ in channel $\gamma$ of conv oper-

---

[2] Cohen et al. (2016b) consider two settings for the $1 \times 1$ conv operator. The first, referred to as *weight sharing*, is the one described above, and corresponds to standard convolution. The second is more general, allowing filters that slide across the previous layer to have different weights at different spatial locations. It is shown in Cohen et al. (2016b) that without weight sharing, a convolutional arithmetic circuit with one hidden layer (or more) is universal, *i.e.* can realize any function if its size (width) is unbounded. This property is imperative for the study of depth efficiency, as that requires shallow networks to ultimately be able to replicate any function realized by a deep network. In this paper we limit the presentation to networks with weight sharing, which are not universal. We do so because they are more conventional, and since our entire analysis is oblivious to whether or not weights are shared (applies as is to both settings). The only exception is where we reproduce the depth efficiency result of Cohen et al. (2016b). There, we momentarily consider networks without weight sharing.

ator at hidden layer 0 is $\{j\}$, and that of an output node is $[N]$, corresponding to the entire input. Denote by $h_{(l,\gamma,j)}$ the function realized by node $j$ of channel $\gamma$ in conv operator at hidden layer $l$, and let $I^{(l,\gamma,j)} \subset [N]$ be its receptive field. By the structure of the network it is evident that $I^{(l,\gamma,j)}$ does not depend on $\gamma$, so we may write $I^{(l,j)}$ instead. Moreover, assuming pooling windows are uniform across channels (as customary with convolutional networks), and taking into account the fact that they do not overlap, we conclude that $I^{(l,j_1)}$ and $I^{(l,j_2)}$ are necessarily disjoint if $j_1 \neq j_2$. A simple induction over $l = 0 \ldots L - 1$ then shows that $h_{(l,\gamma,j)}$ may be expressed as $h_{(l,\gamma,j)}(\mathbf{x}_{i_1}, \ldots, \mathbf{x}_{i_T}) = \sum_{d_1 \ldots d_T = 1}^{M} \mathcal{A}_{d_1 \ldots d_T}^{(l,\gamma,j)} \prod_{t=1}^{T} f_{\theta_{d_t}}(\mathbf{x}_{i_t})$, where $\{i_1, \ldots, i_T\}$ stands for the receptive field $I^{(l,j)}$, and $\mathcal{A}^{(l,\gamma,j)}$ is a tensor of order $T = |I^{(l,j)}|$ and dimension $M$ in each mode, with entries given by polynomials in the network's conv weights $\{\mathbf{a}^{l,\gamma}\}_{l,\gamma}$. Taking the induction one step further (from last hidden layer to network output), we obtain the following expression for functions realized by network outputs:

$$h_y(\mathbf{x}_1, \ldots, \mathbf{x}_N) = \sum_{d_1 \ldots d_N = 1}^{M} \mathcal{A}_{d_1 \ldots d_N}^{y} \prod_{i=1}^{N} f_{\theta_{d_i}}(\mathbf{x}_i) \tag{2}$$

$y \in [Y]$ here is an output node index, and $h_y$ is the function realized by that node. $\mathcal{A}^y$ is a tensor of order $N$ and dimension $M$ in each mode, with entries given by polynomials in the network's conv weights $\{\mathbf{a}^{l,\gamma}\}_{l,\gamma}$ and output weights $\mathbf{a}^{L,y}$. Hereafter, terms such as *function realized by a network* or *coefficient tensor realized by a network*, are to be understood as referring to $h_y$ or $\mathcal{A}^y$ respectively. Next, we present explicit expressions for $\mathcal{A}^y$ under two canonical networks – deep and shallow.

**Deep network.** Consider a network as in fig. 1(a), with pooling windows set to cover four entries each, resulting in $L = \log_4 N$ hidden layers. The linear weights of such a network are $\{\mathbf{a}^{0,\gamma} \in \mathbb{R}^M\}_{\gamma \in [r_0]}$ for conv operator in hidden layer 0, $\{\mathbf{a}^{l,\gamma} \in \mathbb{R}^{r_{l-1}}\}_{\gamma \in [r_l]}$ for conv operator in hidden layer $l = 1 \ldots L - 1$, and $\{\mathbf{a}^{L,y} \in \mathbb{R}^{r_{L-1}}\}_{y \in [Y]}$ for dense output operator. They determine the coefficient tensor $\mathcal{A}^y$ (eq. 2) through the following recursive decomposition:

$$\underbrace{\phi^{1,\gamma}}_{\text{order 4}} = \sum_{\alpha=1}^{r_0} a_\alpha^{1,\gamma} \cdot \otimes^4 \mathbf{a}^{0,\alpha} \qquad , \gamma \in [r_1]$$

$$\cdots$$

$$\underbrace{\phi^{l,\gamma}}_{\text{order } 4^l} = \sum_{\alpha=1}^{r_{l-1}} a_\alpha^{l,\gamma} \cdot \otimes^4 \phi^{l-1,\alpha} \qquad , l \in \{2 \ldots L - 1\}, \gamma \in [r_l]$$

$$\cdots$$

$$\underbrace{\mathcal{A}^y}_{\text{order } 4^L = N} = \sum_{\alpha=1}^{r_{L-1}} a_\alpha^{L,y} \cdot \otimes^4 \phi^{L-1,\alpha} \tag{3}$$

$a_\alpha^{l,\gamma}$ and $a_\alpha^{L,y}$ here are scalars representing entry $\alpha$ in the vectors $\mathbf{a}^{l,\gamma}$ and $\mathbf{a}^{L,y}$ respectively, and the symbol $\otimes$ with a superscript stands for a repeated tensor product, *e.g.* $\otimes^4 \mathbf{a}^{0,\alpha} := \mathbf{a}^{0,\alpha} \otimes \mathbf{a}^{0,\alpha} \otimes \mathbf{a}^{0,\alpha} \otimes \mathbf{a}^{0,\alpha}$. To verify that under pooling windows of size four $\mathcal{A}^y$ is indeed given by eq. 3, simply plug the rows of the decomposition into eq. 2, starting from bottom and continuing upwards. For context, eq. 3 describes what is known as a hierarchical tensor decomposition (see chapter 11 in Hackbusch (2012)), with underlying tree over modes being a full quad-tree (corresponding to the fact that the network's pooling windows cover four entries each).

**Shallow network.** The second network we pay special attention to is shallow, comprising a single hidden layer with global pooling – see illustration in fig. 1(b). The linear weights of such a network are $\{\mathbf{a}^{0,\gamma} \in \mathbb{R}^M\}_{\gamma \in [r_0]}$ for hidden conv operator and $\{\mathbf{a}^{1,y} \in \mathbb{R}^{r_0}\}_{y \in [Y]}$ for dense output operator. They determine the coefficient tensor $\mathcal{A}^y$ (eq. 2) as follows:

$$\mathcal{A}^y = \sum_{\gamma=1}^{r_0} a_\gamma^{1,y} \cdot \otimes^N \mathbf{a}^{0,\gamma} \tag{4}$$

where $a_\gamma^{1,y}$ stands for entry $\gamma$ of $\mathbf{a}^{1,y}$, and again, the symbol $\otimes$ with a superscript represents a repeated tensor product. The tensor decomposition in eq. 4 is an instance of the classic CP decomposition, also known as rank-1 decomposition (see Kolda and Bader (2009) for a historic survey).

To conclude this section, we relate the background material above, as well as our contribution described in the upcoming sections, to the work of Cohen et al. (2016b). The latter shows that with

arbitrary coefficient tensors $\mathcal{A}^y$, functions $h_y$ as in eq. 2 form a universal hypotheses space. It is then shown that convolutional arithmetic circuits as in fig. 1(a) realize such functions by applying tensor decompositions to $\mathcal{A}^y$, with the type of decomposition determined by the structure of a network (number of layers, number of channels in each layer *etc.*). The deep network (fig. 1(a) with size-4 pooling windows and $L = \log_4 N$ hidden layers) and the shallow network (fig. 1(b)) presented here-inabove are two special cases, whose corresponding tensor decompositions are given in eq. 3 and 4 respectively. The central result in Cohen et al. (2016b) relates to inductive bias through the notion of depth efficiency – it is shown that in the parameter space of a deep network, all weight settings but a set of (Lebesgue) measure zero give rise to functions that can only be realized (or approximated) by a shallow network if the latter has exponential size. This result does not relate to the characteristics of instances $X = (\mathbf{x}_1, \ldots, \mathbf{x}_N)$, it only treats the ability of shallow networks to replicate functions realized by deep networks.

In this paper we draw a line connecting the inductive bias to the nature of $X$, by studying the relation between a network's architecture and its ability to model correlation among patches $\mathbf{x}_i$. Specifically, in sec. 4 we consider partitions $(I, J)$ of $[N]$ ($I \dot\cup J = [N]$, where $\dot\cup$ stands for disjoint union), and present the notion of separation rank as a measure of the correlation modeled between the patches indexed by $I$ and those indexed by $J$. In sec. 5.1 the separation rank of a network's function $h_y$ w.r.t. a partition $(I, J)$ is proven to be equal to the rank of $[\![\mathcal{A}^y]\!]_{I,J}$ – the matricization of the coefficient tensor $\mathcal{A}^y$ w.r.t. $(I, J)$. Sec. 5.2 derives lower and upper bounds on this rank for a deep network, showing that it supports exponential separation ranks with polynomial size for certain partitions, whereas for others it is required to be exponentially large. Subsequently, sec. 5.3 establishes an upper bound on $rank[\![\mathcal{A}^y]\!]_{I,J}$ for shallow networks, implying that these must be exponentially large in order to model exponential separation rank under any partition, and thus cannot efficiently replicate a deep network's correlations. Our analysis concludes in sec. 6, where we discuss the pooling geometry of a deep network as a means for controlling the inductive bias by determining a correspondence between partitions $(I, J)$ and spatial partitions of the input. Finally, we demonstrate experimentally in sec. 7 how different pooling geometries lead to superior performance in different tasks. Our experiments include not only convolutional arithmetic circuits, but also convolutional rectifier networks, *i.e.* convolutional networks with ReLU activation and max or average pooling.

## 4 SEPARATION RANK

In this section we define the concept of separation rank for functions realized by convolutional arithmetic circuits (sec. 3), *i.e.* real functions that take as input $X = (\mathbf{x}_1, \ldots, \mathbf{x}_N) \in (\mathbb{R}^s)^N$. The separation rank serves as a measure of the correlations such functions induce between different sets of input patches, *i.e.* different subsets of the variable set $\{\mathbf{x}_1, \ldots, \mathbf{x}_N\}$.

Let $(I, J)$ be a partition of input indexes, *i.e.* $I$ and $J$ are disjoint subsets of $[N]$ whose union gives $[N]$. We may write $I = \{i_1, \ldots, i_{|I|}\}$ where $i_1 < \cdots < i_{|I|}$, and similarly $J = \{j_1, \ldots, j_{|J|}\}$ where $j_1 < \cdots < j_{|J|}$. For a function $h : (\mathbb{R}^s)^N \to \mathbb{R}$, the *separation rank w.r.t. the partition $(I, J)$* is defined as follows: [3]

$$sep(h; I, J) := \min \Big\{ R \in \mathbb{N} \cup \{0\} : \exists g_1 \ldots g_R : (\mathbb{R}^s)^{|I|} \to \mathbb{R}, g'_1 \ldots g'_R : (\mathbb{R}^s)^{|J|} \to \mathbb{R} \ s.t. \quad (5)$$

$$h(\mathbf{x}_1, \ldots, \mathbf{x}_N) = \sum\nolimits_{\nu=1}^{R} g_\nu(\mathbf{x}_{i_1}, \ldots, \mathbf{x}_{i_{|I|}}) g'_\nu(\mathbf{x}_{j_1}, \ldots, \mathbf{x}_{j_{|J|}}) \Big\}$$

In words, it is the minimal number of summands that together give $h$, where each summand is *separable w.r.t. $(I, J)$*, *i.e.* is equal to a product of two functions – one that intakes only patches indexed by $I$, and another that intakes only patches indexed by $J$. One may wonder if it is at all possible to express $h$ through such summands, *i.e.* if the separation rank of $h$ is finite. From the theory of tensor products between $L^2$ spaces (see Hackbusch (2012) for a comprehensive coverage), we know that any $h \in L^2((\mathbb{R}^s)^N)$, *i.e.* any $h$ that is measurable and square-integrable, may be approximated arbitrarily well by summations of the form $\sum_{\nu=1}^{R} g_\nu(\mathbf{x}_{i_1}, \ldots, \mathbf{x}_{i_{|I|}}) g'_\nu(\mathbf{x}_{j_1}, \ldots, \mathbf{x}_{j_{|J|}})$. Exact realization however is only guaranteed at the limit $R \to \infty$, thus in general the separation rank of $h$

---

[3] If $I = \emptyset$ or $J = \emptyset$ then by definition $sep(h; I, J) = 1$ (unless $h \equiv 0$, in which case $sep(h; I, J) = 0$).

need not be finite. Nonetheless, as we show in sec. 5, for the class of functions we are interested in, namely functions realizable by convolutional arithmetic circuits, separation ranks are always finite.

The concept of separation rank was introduced in Beylkin and Mohlenkamp (2002) for numerical treatment of high-dimensional functions, and has since been employed for various applications, *e.g.* quantum chemistry (Harrison et al. (2003)), particle engineering (Hackbusch (2006)) and machine learning (Beylkin et al. (2009)). If the separation rank of a function w.r.t. a partition of its input is equal to 1, the function is separable, meaning it does not model any interaction between the sets of variables. Specifically, if $sep(h; I, J) = 1$ then there exist $g : (\mathbb{R}^s)^{|I|} \to \mathbb{R}$ and $g' : (\mathbb{R}^s)^{|J|} \to \mathbb{R}$ such that $h(\mathbf{x}_1, \ldots, \mathbf{x}_N) = g(\mathbf{x}_{i_1}, \ldots, \mathbf{x}_{i_{|I|}}) g'(\mathbf{x}_{j_1}, \ldots, \mathbf{x}_{j_{|J|}})$, and the function $h$ cannot take into account consistency between the values of $\{\mathbf{x}_{i_1}, \ldots, \mathbf{x}_{i_{|I|}}\}$ and those of $\{\mathbf{x}_{j_1}, \ldots, \mathbf{x}_{j_{|J|}}\}$. In a statistical setting, if $h$ is a probability density function, this would mean that $\{\mathbf{x}_{i_1}, \ldots, \mathbf{x}_{i_{|I|}}\}$ and $\{\mathbf{x}_{j_1}, \ldots, \mathbf{x}_{j_{|J|}}\}$ are statistically independent. The higher $sep(h; I, J)$ is, the farther $h$ is from this situation, *i.e.* the more it models dependency between $\{\mathbf{x}_{i_1}, \ldots, \mathbf{x}_{i_{|I|}}\}$ and $\{\mathbf{x}_{j_1}, \ldots, \mathbf{x}_{j_{|J|}}\}$, or equivalently, the stronger the correlation it induces between the patches indexed by $I$ and those indexed by $J$.

The interpretation of separation rank as a measure of deviation from separability is formalized in app. B, where it is shown that $sep(h; I, J)$ is closely related to the $L^2$ distance of $h$ from the set of separable functions w.r.t. $(I, J)$. Specifically, we define $D(h; I, J)$ as the latter distance divided by the $L^2$ norm of $h$ [4], and show that $sep(h; I, J)$ provides an upper bound on $D(h; I, J)$. While it is not possible to lay out a general lower bound on $D(h; I, J)$ in terms of $sep(h; I, J)$, we show that the specific lower bounds on $sep(h; I, J)$ underlying our analyses can be translated into lower bounds on $D(h; I, J)$. This implies that our results, facilitated by upper and lower bounds on separation ranks of convolutional arithmetic circuits, may equivalently be framed in terms of $L^2$ distances from separable functions.

## 5 CORRELATION ANALYSIS

In this section we analyze convolutional arithmetic circuits (sec. 3) in terms of the correlations they can model between sides of different input partitions, *i.e.* in terms of the separation ranks (sec. 4) they support under different partitions $(I, J)$ of $[N]$. We begin in sec. 5.1, establishing a correspondence between separation ranks and coefficient tensor matricization ranks. This correspondence is then used in sec. 5.2 and 5.3 to analyze the deep and shallow networks (respectively) presented in sec. 3. We note that we focus on these particular networks merely for simplicity of presentation – the analysis can easily be adapted to account for alternative networks with different depths and pooling schemes.

### 5.1 FROM SEPARATION RANK TO MATRICIZATION RANK

Let $h_y$ be a function realized by a convolutional arithmetic circuit, with corresponding coefficient tensor $\mathcal{A}^y$ (eq. 2). Denote by $(I, J)$ an arbitrary partition of $[N]$, *i.e.* $I \dot\cup J = [N]$. We are interested in studying $sep(h_y; I, J)$ – the separation rank of $h_y$ w.r.t. $(I, J)$ (eq. 5). As claim 1 below states, assuming representation functions $\{f_{\theta_d}\}_{d \in [M]}$ are linearly independent (if they are not, we drop dependent functions and modify $\mathcal{A}^y$ accordingly [5]), this separation rank is equal to the rank of $[\![\mathcal{A}^y]\!]_{I,J}$ – the matricization of the coefficient tensor $\mathcal{A}^y$ w.r.t. the partition $(I, J)$. Our problem thus translates to studying ranks of matricized coefficient tensors.

**Claim 1.** *Let $h_y$ be a function realized by a convolutional arithmetic circuit (fig. 1(a)), with corresponding coefficient tensor $\mathcal{A}^y$ (eq. 2). Assume that the network's representation functions $f_{\theta_d}$ are linearly independent, and that they, as well as the functions $g_\nu, g'_\nu$ in the definition of separation*

---

[4] The normalization (division by norm) is of critical importance – without it rescaling $h$ would accordingly rescale $D(h; I, J)$, rendering the latter uninformative in terms of deviation from separability.

[5] Suppose for example that $f_{\theta_M}$ is dependent, *i.e.* there exist $\alpha_1 \ldots \alpha_{M-1} \in \mathbb{R}$ such that $f_{\theta_M}(\mathbf{x}) = \sum_{d=1}^{M-1} \alpha_d \cdot f_{\theta_d}(\mathbf{x})$. We may then plug this into eq. 2, and obtain an expression for $h_y$ that has $f_{\theta_1} \ldots f_{\theta_{M-1}}$ as representation functions, and a coefficient tensor with dimension $M - 1$ in each mode. Continuing in this fashion, one arrives at an expression for $h_y$ whose representation functions are linearly independent.

*rank (eq. 5), are measurable and square-integrable.* [6] *Then, for any partition $(I, J)$ of $[N]$, it holds that $sep(h_y; I, J) = rank[\![\mathcal{A}^y]\!]_{I,J}$.*

*Proof.* See app. A.1. □

As the linear weights of a network vary, so do the coefficient tensors ($\mathcal{A}^y$) it gives rise to. Accordingly, for a particular partition $(I, J)$, a network does not correspond to a single value of $rank[\![\mathcal{A}^y]\!]_{I,J}$, but rather supports a range of values. We analyze this range by quantifying its maximum, which reflects the strongest correlation that the network can model between the input patches indexed by $I$ and those indexed by $J$. One may wonder if the maximal value of $rank[\![\mathcal{A}^y]\!]_{I,J}$ is the appropriate statistic to measure, as a-priori, it may be that $rank[\![\mathcal{A}^y]\!]_{I,J}$ is maximal for very few of the network's weight settings, and much lower for all the rest. Apparently, as claim 2 below states, this is not the case, and in fact $rank[\![\mathcal{A}^y]\!]_{I,J}$ is maximal under almost all of the network's weight settings.

**Claim 2.** *Consider a convolutional arithmetic circuit (fig. 1(a)) with corresponding coefficient tensor $\mathcal{A}^y$ (eq. 2). $\mathcal{A}^y$ depends on the network's linear weights – $\{\mathbf{a}^{l,\gamma}\}_{l,\gamma}$ and $\mathbf{a}^{L,y}$, thus for a given partition $(I, J)$ of $[N]$, $rank[\![\mathcal{A}^y]\!]_{I,J}$ is a function of these weights. This function obtains its maximum almost everywhere (w.r.t. Lebesgue measure).*

*Proof.* See app. A.2. □

## 5.2 DEEP NETWORK

In this subsection we study correlations modeled by the deep network presented in sec. 3 (fig. 1(a) with size-4 pooling windows and $L = \log_4 N$ hidden layers). In accordance with sec. 5.1, we do so by characterizing the maximal ranks of coefficient tensor matricizations under different partitions.

Recall from eq. 3 the hierarchical decomposition expressing a coefficient tensor $\mathcal{A}^y$ realized by the deep network. We are interested in matricizations of this tensor under different partitions of $[N]$. Let $(I, J)$ be an arbitrary partition, *i.e.* $I\dot\cup J = [N]$. Matricizing the last level of eq. 3 w.r.t. $(I, J)$, while applying the relation in eq. 1, gives:

$$
\begin{aligned}
[\![\mathcal{A}^y]\!]_{I,J} &= \sum_{\alpha=1}^{r_{L-1}} a_\alpha^{L,y} \cdot [\![\phi^{L-1,\alpha} \otimes \phi^{L-1,\alpha} \otimes \phi^{L-1,\alpha} \otimes \phi^{L-1,\alpha}]\!]_{I,J} \\
&= \sum_{\alpha=1}^{r_{L-1}} a_\alpha^{L,y} \cdot [\![\phi^{L-1,\alpha} \otimes \phi^{L-1,\alpha}]\!]_{I\cap[2\cdot4^{L-1}],J\cap[2\cdot4^{L-1}]} \\
&\qquad \odot [\![\phi^{L-1,\alpha} \otimes \phi^{L-1,\alpha}]\!]_{(I-2\cdot4^{L-1})\cap[2\cdot4^{L-1}],(J-2\cdot4^{L-1})\cap[2\cdot4^{L-1}]}
\end{aligned}
$$

Applying eq. 1 again, this time to matricizations of the tensor $\phi^{L-1,\alpha} \otimes \phi^{L-1,\alpha}$, we obtain:

$$
\begin{aligned}
[\![\mathcal{A}^y]\!]_{I,J} &= \sum_{\alpha=1}^{r_{L-1}} a_\alpha^{L,y} \cdot [\![\phi^{L-1,\alpha}]\!]_{I\cap[4^{L-1}],J\cap[4^{L-1}]} \\
&\qquad \odot [\![\phi^{L-1,\alpha}]\!]_{(I-4^{L-1})\cap[4^{L-1}],(J-4^{L-1})\cap[4^{L-1}]} \\
&\qquad \odot [\![\phi^{L-1,\alpha}]\!]_{(I-2\cdot4^{L-1})\cap[4^{L-1}],(J-2\cdot4^{L-1})\cap[4^{L-1}]} \\
&\qquad \odot [\![\phi^{L-1,\alpha}]\!]_{(I-3\cdot4^{L-1})\cap[4^{L-1}],(J-3\cdot4^{L-1})\cap[4^{L-1}]}
\end{aligned}
$$

For every $k \in [4]$ define $I_{L-1,k} := (I-(k-1)\cdot4^{L-1})\cap[4^{L-1}]$ and $J_{L-1,k} := (J-(k-1)\cdot4^{L-1})\cap[4^{L-1}]$. In words, $(I_{L-1,k}, J_{L-1,k})$ represents the partition induced by $(I, J)$ on the $k$'th quadrant of $[N]$, *i.e.* on the $k$'th size-$4^{L-1}$ group of input patches. We now have the following matricized version of the last level in eq. 3:

$$
[\![\mathcal{A}^y]\!]_{I,J} = \sum_{\alpha=1}^{r_{L-1}} a_\alpha^{L,y} \cdot \overset{4}{\underset{t=1}{\odot}} [\![\phi^{L-1,\alpha}]\!]_{I_{L-1,t},J_{L-1,t}}
$$

---

[6] Square-integrability of representation functions $f_{\theta_d}$ may seem as a limitation at first glance, as for example neurons $f_{\theta_d}(\mathbf{x}) = \sigma(\mathbf{w}_d^\top \mathbf{x} + b_d)$, with parameters $\theta_d = (\mathbf{w}_d, b_d) \in \mathbb{R}^s \times \mathbb{R}$ and sigmoid or ReLU activation $\sigma(\cdot)$, do not meet this condition. However, since in practice our inputs are bounded (*e.g.* they represent image pixels by holding intensity values), we may view functions as having compact support, which, as long as they are continuous (holds in all cases of interest), ensures square-integrability.

where the symbol $\odot$ with a running index stands for an iterative Kronecker product. To derive analogous matricized versions for the upper levels of eq. 3, we define for $l \in \{0 \ldots L-1\}, k \in [N/4^l]$:

$$I_{l,k} := (I - (k-1) \cdot 4^l) \cap [4^l] \qquad J_{l,k} := (J - (k-1) \cdot 4^l) \cap [4^l] \qquad (6)$$

That is to say, $(I_{l,k}, J_{l,k})$ represents the partition induced by $(I, J)$ on the set of indexes $\{(k-1) \cdot 4^l + 1, \ldots, k \cdot 4^l\}$, *i.e.* on the $k$'th size-$4^l$ group of input patches. With this notation in hand, traversing upwards through the levels of eq. 3, with repeated application of the relation in eq. 1, one arrives at the following matrix decomposition for $[\![\mathcal{A}^y]\!]_{I,J}$:

$$\underbrace{[\![\phi^{1,\gamma}]\!]_{I_{1,k},J_{1,k}}}_{M^{|I_{1,k}|}\text{-by-}M^{|J_{1,k}|}} = \sum_{\alpha=1}^{r_0} a_\alpha^{1,\gamma} \cdot \underset{t=1}{\overset{4}{\odot}} [\![\mathbf{a}^{0,\alpha}]\!]_{I_{0,4(k-1)+t},J_{0,4(k-1)+t}} \qquad , \gamma \in [r_1]$$

$$\cdots$$

$$\underbrace{[\![\phi^{l,\gamma}]\!]_{I_{l,k},J_{l,k}}}_{M^{|I_{l,k}|}\text{-by-}M^{|J_{l,k}|}} = \sum_{\alpha=1}^{r_{l-1}} a_\alpha^{l,\gamma} \cdot \underset{t=1}{\overset{4}{\odot}} [\![\phi^{l-1,\alpha}]\!]_{I_{l-1,4(k-1)+t},J_{l-1,4(k-1)+t}} \quad , l \in \{2 \ldots L-1\}, \gamma \in [r_l]$$

$$\cdots$$

$$\underbrace{[\![\mathcal{A}^y]\!]_{I,J}}_{M^{|I|}\text{-by-}M^{|J|}} = \sum_{\alpha=1}^{r_{L-1}} a_\alpha^{L,y} \cdot \underset{t=1}{\overset{4}{\odot}} [\![\phi^{L-1,\alpha}]\!]_{I_{L-1,t},J_{L-1,t}} \qquad (7)$$

Eq. 7 expresses $[\![\mathcal{A}^y]\!]_{I,J}$ – the matricization w.r.t. the partition $(I, J)$ of a coefficient tensor $\mathcal{A}^y$ realized by the deep network, in terms of the network's conv weights $\{\mathbf{a}^{l,\gamma}\}_{l,\gamma}$ and output weights $\mathbf{a}^{L,y}$. As discussed above, our interest lies in the maximal rank that this matricization can take. Theorem 1 below provides lower and upper bounds on this maximal rank, by making use of eq. 7, and of the rank-multiplicative property of the Kronecker product ($rank(A \odot B) = rank(A) \cdot rank(B)$).

**Theorem 1.** *Let $(I, J)$ be a partition of $[N]$, and $[\![\mathcal{A}^y]\!]_{I,J}$ be the matricization w.r.t. $(I, J)$ of a coefficient tensor $\mathcal{A}^y$ (eq. 2) realized by the deep network (fig. 1(a) with size-4 pooling windows). For every $l \in \{0 \ldots L-1\}$ and $k \in [N/4^l]$, define $I_{l,k}$ and $J_{l,k}$ as in eq. 6. Then, the maximal rank that $[\![\mathcal{A}^y]\!]_{I,J}$ can take (when network weights vary) is:*

- *No smaller than $\min\{r_0, M\}^S$, where $S := |\{k \in [N/4] : I_{1,k} \neq \emptyset \wedge J_{1,k} \neq \emptyset\}|$.*

- *No greater than $\min\{M^{\min\{|I|,|J|\}}, r_{L-1} \prod_{t=1}^4 c^{L-1,t}\}$, where $c^{0,k} := 1$ for $k \in [N]$, and $c^{l,k} := \min\{M^{\min\{|I_{l,k}|,|J_{l,k}|\}}, r_{l-1} \prod_{t=1}^4 c^{l-1,4(k-1)+t}\}$ for $l \in [L-1], k \in [N/4^l]$.*

*Proof.* See app. A.3. □

The lower bound in theorem 1 is exponential in $S$, the latter defined to be the number of size-4 patch groups that are split by the partition $(I, J)$, *i.e.* whose indexes are divided between $I$ and $J$. Partitions that split many of the size-4 patch groups will thus lead to a large lower bound. For example, consider the partition $(I^{odd}, J^{even})$ defined as follows:

$$I^{odd} = \{1, 3, \ldots, N-1\} \qquad J^{even} = \{2, 4, \ldots, N\} \qquad (8)$$

This partition splits all size-4 patch groups ($S = N/4$), leading to a lower bound that is exponential in the number of patches ($N$).

The upper bound in theorem 1 is expressed via constants $c^{l,k}$, defined recursively over levels $l = 0 \ldots L-1$, with $k$ ranging over $1 \ldots N/4^l$ for each level $l$. What prevents $c^{l,k}$ from growing double-exponentially fast (w.r.t. $l$) is the minimization with $M^{\min\{|I_{l,k}|,|J_{l,k}|\}}$. Specifically, if $\min\{|I_{l,k}|, |J_{l,k}|\}$ is small, *i.e.* if the partition induced by $(I, J)$ on the $k$'th size-$4^l$ group of patches is unbalanced (most of the patches belong to one side of the partition, and only a few belong to the other), $c^{l,k}$ will be of reasonable size. The higher this takes place in the hierarchy (*i.e.* the larger $l$ is), the lower our eventual upper bound will be. In other words, if partitions induced by $(I, J)$ on size-$4^l$ patch groups are unbalanced for large values of $l$, the upper bound in theorem 1 will be small. For example, consider the partition $(I^{low}, J^{high})$ defined by:

$$I^{low} = \{1, \ldots, N/2\} \qquad J^{high} = \{N/2 + 1, \ldots, N\} \qquad (9)$$

Under $(I^{low}, J^{high})$, all partitions induced on size-$4^{L-1}$ patch groups (quadrants of $[N]$) are completely one-sided ($\min\{|I_{L-1,k}|, |J_{L-1,k}|\} = 0$ for all $k \in [4]$), resulting in the upper bound being no greater than $r_{L-1}$ – linear in network size.

To summarize this discussion, theorem 1 states that with the deep network, the maximal rank of a coefficient tensor matricization w.r.t. $(I, J)$, highly depends on the nature of the partition $(I, J)$ – it will be exponentially high for partitions such as $(I^{odd}, J^{even})$, that split many size-4 patch groups, while being only polynomial (or linear) for partitions like $(I^{low}, J^{high})$, under which size-$4^l$ patch groups are unevenly divided for large values of $l$. Since the rank of a coefficient tensor matricization w.r.t. $(I, J)$ corresponds to the strength of correlation modeled between input patches indexed by $I$ and those indexed by $J$ (sec. 5.1), we conclude that the ability of a polynomially sized deep network to model correlation between sets of input patches highly depends on the nature of these sets.

## 5.3 SHALLOW NETWORK

We now turn to study correlations modeled by the shallow network presented in sec. 3 (fig. 1(b)). In line with sec. 5.1, this is achieved by characterizing the maximal ranks of coefficient tensor matricizations under different partitions.

Recall from eq. 4 the CP decomposition expressing a coefficient tensor $\mathcal{A}^y$ realized by the shallow network. For an arbitrary partition $(I, J)$ of $[N]$, *i.e.* $I \dot\cup J = [N]$, matricizing this decomposition with repeated application of the relation in eq. 1, gives the following expression for $[\![\mathcal{A}^y]\!]_{I,J}$ – the matricization w.r.t. $(I, J)$ of a coefficient tensor realized by the shallow network:

$$[\![\mathcal{A}^y]\!]_{I,J} = \sum\nolimits_{\gamma=1}^{r_0} a_\gamma^{1,y} \cdot \left( \odot^{|I|} \mathbf{a}^{0,\gamma} \right) \left( \odot^{|J|} \mathbf{a}^{0,\gamma} \right)^\top \tag{10}$$

$\odot^{|I|}\mathbf{a}^{0,\gamma}$ and $\odot^{|J|}\mathbf{a}^{0,\gamma}$ here are column vectors of dimensions $M^{|I|}$ and $M^{|J|}$ respectively, standing for the Kronecker products of $\mathbf{a}^{0,\gamma} \in \mathbb{R}^M$ with itself $|I|$ and $|J|$ times (respectively). Eq. 10 immediately leads to two observations regarding the ranks that may be taken by $[\![\mathcal{A}^y]\!]_{I,J}$. First, they depend on the partition $(I, J)$ only through its division size, *i.e.* through $|I|$ and $|J|$. Second, they are no greater than $\min\{M^{\min\{|I|,|J|\}}, r_0\}$, meaning that the maximal rank is linear (or less) in network size. In light of sec. 5.1 and 5.2, these findings imply that in contrast to the deep network, which with polynomial size supports exponential separation ranks under favored partitions, the shallow network treats all partitions (of a given division size) equally, and can only give rise to an exponential separation rank if its size is exponential.

Suppose now that we would like to use the shallow network to replicate a function realized by a polynomially sized deep network. So long as the deep network's function admits an exponential separation rank under at least one of the favored partitions (*e.g.* $(I^{odd}, J^{even})$ – eq. 8), the shallow network would have to be exponentially large in order to replicate it, *i.e.* depth efficiency takes place. [7] Since all but a negligible set of the functions realizable by the deep network give rise to maximal separation ranks (sec 5.1), we obtain the complete depth efficiency result of Cohen et al. (2016b). However, unlike Cohen et al. (2016b), which did not provide any explanation for the usefulness of functions brought forth by depth, we obtain an insight into their utility – they are able to efficiently model strong correlation under favored partitions of the input.

## 6 INDUCTIVE BIAS THROUGH POOLING GEOMETRY

The deep network presented in sec. 3, whose correlations we analyzed in sec. 5.2, was defined as having size-4 pooling windows, *i.e.* pooling windows covering four entries each. We have yet

---

[7] Convolutional arithmetic circuits as we have defined them (sec. 3) are not universal. In particular, it may very well be that a function realized by a polynomially sized deep network cannot be replicated by the shallow network, no matter how large (wide) we allow it to be. In such scenarios depth efficiency does not provide insight into the complexity of functions brought forth by depth. To obtain a shallow network that is universal, thus an appropriate gauge for depth efficiency, we may remove the constraint of weight sharing, *i.e.* allow the filters in the hidden conv operator to hold different weights at different spatial locations (see Cohen et al. (2016b) for proof that this indeed leads to universality). All results we have established for the original shallow network remain valid when weight sharing is removed. In particular, the separation ranks of the network are still linear in its size. This implies that as suggested, depth efficiency indeed holds.

to specify the shapes of these windows, or equivalently, the spatial (two-dimensional) locations of nodes grouped together in the process of pooling. In compliance with standard convolutional network design, we now assume that the network's (size-4) pooling windows are contiguous square blocks, *i.e.* have shape $2 \times 2$. Under this configuration, the network's functional description (eq. 2 with $\mathcal{A}^y$ given by eq. 3) induces a spatial ordering of input patches [8] , which may be described by the following recursive process:

- Set the index of the top-left patch to $1$.

- For $l = 1, \ldots, L = log_4 N$: Replicate the already-assigned top-left $2^{l-1}$-by-$2^{l-1}$ block of indexes, and place copies on its right, bottom-right and bottom. Then, add a $4^{l-1}$ offset to all indexes in the right copy, a $2 \cdot 4^{l-1}$ offset to all indexes in the bottom-right copy, and a $3 \cdot 4^{l-1}$ offset to all indexes in the bottom copy.

With this spatial ordering (illustrated in fig. 1(c)), partitions $(I, J)$ of $[N]$ convey a spatial pattern. For example, the partition $(I^{odd}, J^{even})$ (eq. 8) corresponds to the pattern illustrated on the left of fig. 1(c), whereas $(I^{low}, J^{high})$ (eq. 9) corresponds to the pattern illustrated on the right. Our analysis (sec. 5.2) shows that the deep network is able to model strong correlation under $(I^{odd}, J^{even})$, while being inefficient for modeling correlation under $(I^{low}, J^{high})$. More generally, partitions for which $S$, defined in theorem 1, is high, convey patterns that split many $2 \times 2$ patch blocks, *i.e.* are highly entangled. These partitions enjoy the possibility of strong correlation. On the other hand, partitions for which $\min\{|I_{l,k}|, |J_{l,k}|\}$ is small for large values of $l$ (see eq. 6 for definition of $I_{l,k}$ and $J_{l,k}$) convey patterns that divide large $2^l \times 2^l$ patch blocks unevenly, *i.e.* separate the input to distinct contiguous regions. These partitions, as we have seen, suffer from limited low correlations.

We conclude that with $2 \times 2$ pooling, the deep network is able to model strong correlation between input regions that are highly entangled, at the expense of being inefficient for modeling correlation between input regions that are far apart. Had we selected a different pooling regime, the preference of input partition patterns in terms of modeled correlation would change. For example, if pooling windows were set to group nodes with their spatial reflections (horizontal, vertical and horizontal-vertical), coarse patterns that divide the input symmetrically, such as the one illustrated on the right of fig. 1(c), would enjoy the possibility of strong correlation, whereas many entangled patterns would now suffer from limited low correlation. The choice of pooling shapes thus serves as a means for controlling the inductive bias in terms of correlations modeled between input regions. Square contiguous windows, as commonly employed in practice, lead to a preference that complies with our intuition regarding the statistics of natural images (nearby pixels more correlated than distant ones). Other pooling schemes lead to different preferences, and this allows tailoring a network to data that departs from the usual domain of natural imagery. We demonstrate this experimentally in the next section, where it is shown how different pooling geometries lead to superior performance in different tasks.

## 7 EXPERIMENTS

The main conclusion from our analyses (sec. 5 and 6) is that the pooling geometry of a deep convolutional network controls its inductive bias by determining which correlations between input regions can be modeled efficiently. We have also seen that shallow networks cannot model correlations efficiently, regardless of the considered input regions. In this section we validate these assertions empirically, not only with convolutional arithmetic circuits (subject of our analyses), but also with convolutional rectifier networks – convolutional networks with ReLU activation and max or average pooling. For conciseness, we defer to app. C some details regarding our implementation. The latter is fully available online at `https://github.com/HUJI-Deep/inductive-pooling`.

---

[8] The network's functional description assumes a one-dimensional full quad-tree grouping of input patch indexes. That is to say, it assumes that in the first pooling operation (hidden layer 0), the nodes corresponding to patches $x_1, x_2, x_3, x_4$ are pooled into one group, those corresponding to $x_5, x_6, x_7, x_8$ are pooled into another, and so forth. Similar assumptions hold for the deeper layers. For example, in the second pooling operation (hidden layer 1), the node with receptive field $\{1, 2, 3, 4\}$, *i.e.* the one corresponding to the quadruple of patches $\{x_1, x_2, x_3, x_4\}$, is assumed to be pooled together with the nodes whose receptive fields are $\{5, 6, 7, 8\}$, $\{9, 10, 11, 12\}$ and $\{13, 14, 15, 16\}$.

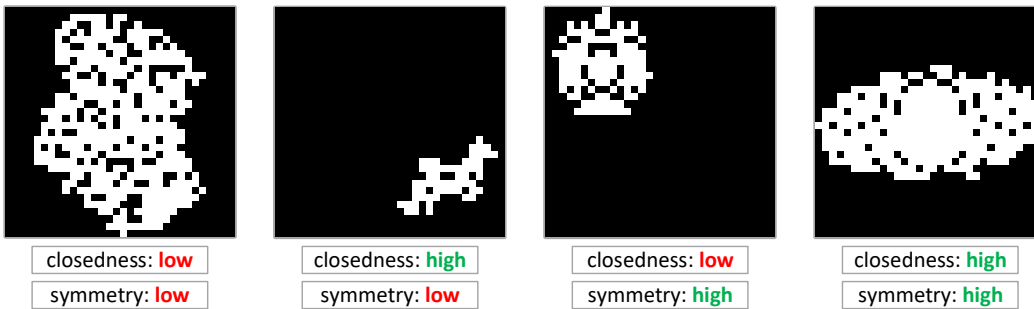

Figure 2: Sample of images from our synthetic classification benchmark. Each image displays a random blob with holes, whose morphological closure and left-right symmetry about its center are measured. Two classification tasks are defined – one for closedness and one for symmetry. In each task, the objective is to distinguish between blobs whose respective property (closedness/symmetry) is high, and ones for which it is low. The tasks differ in nature – closedness requires modeling correlations between neighboring pixels, whereas symmetry requires modeling correlations between pixels and their reflections.

Our experiments are based on a synthetic classification benchmark inspired by medical imaging tasks. Instances to be classified are 32-by-32 binary images, each displaying a random distorted oval shape (*blob*) with missing pixels in its interior (*holes*). For each image, two continuous scores in range $[0, 1]$ are computed. The first, referred to as *closedness*, reflects how morphologically closed a blob is, and is defined to be the ratio between the number of pixels in the blob, and the number of pixels in its closure (see app. D for exact definition of the latter). The second score, named *symmetry*, reflects the degree to which a blob is left-right symmetric about its center. It is measured by cropping the bounding box around a blob, applying a left-right flip to the latter, and computing the ratio between the number of pixels in the intersection of the blob and its reflection, and the number of pixels in the blob. To generate labeled sets for classification (train and test), we render multiple images, sort them according to their closedness and symmetry, and for each of the two scores, assign the label "high" to the top 40% and the label "low" to the bottom 40% (the mid 20% are considered ill-defined). This creates two binary (two-class) classification tasks – one for closedness and one for symmetry (see fig. 2 for a sample of images participating in both tasks). Given that closedness is a property of a local nature, we expect its classification task to require a predictor to be able to model strong correlations between neighboring pixels. Symmetry on the other hand is a property that relates pixels to their reflections, thus we expect its classification task to demand that a predictor be able to model correlations across distances.

We evaluated the deep convolutional arithmetic circuit considered throughout the paper (fig. 1(a) with size-$4$ pooling windows) under two different pooling geometries. The first, referred to as *square*, comprises standard $2 \times 2$ pooling windows. The second, dubbed *mirror*, pools together nodes with their horizontal, vertical and horizontal-vertical reflections. In both cases, input patches ($\mathbf{x}_i$) were set as individual pixels, resulting in $N = 1024$ patches and $L = \log_4 N = 5$ hidden layers. $M = 2$ representation functions ($f_{\theta_d}$) were fixed, the first realizing the identity on binary inputs ($f_{\theta_1}(b) = b$ for $b \in \{0, 1\}$), and the second realizing negation ($f_{\theta_2}(b) = 1 - b$ for $b \in \{0, 1\}$). Classification was realized through $Y = 2$ network outputs, with prediction following the stronger activation. The number of channels across all hidden layers was uniform, and varied between $8$ and $128$. Fig. 3 shows the results of applying the deep network with both square and mirror pooling, to both closedness and symmetry tasks, where each of the latter has $20000$ images for training and $4000$ images for testing. As can be seen in the figure, square pooling significantly outperforms mirror pooling in closedness classification, whereas the opposite occurs in symmetry classification. This complies with our discussion in sec. 6, according to which square pooling supports modeling correlations between entangled (neighboring) regions of the input, whereas mirror pooling puts focus on correlations between input regions that are symmetric w.r.t. one another. We thus obtain a demonstration of how prior knowledge regarding a task at hand may be used to tailor the inductive bias of a deep convolutional network by designing an appropriate pooling geometry.

In addition to the deep network, we also evaluated the shallow convolutional arithmetic circuit analyzed in the paper (fig. 1(b)). The architectural choices for this network were the same as those

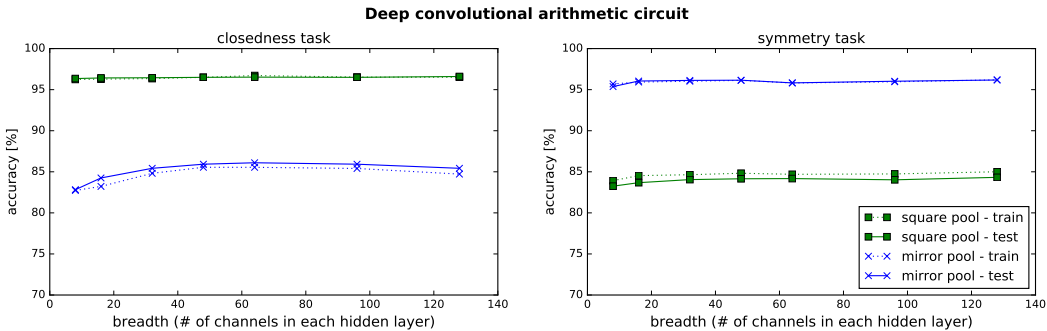

Figure 3: Results of applying a deep convolutional arithmetic circuit to closedness and symmetry classification tasks. Two pooling geometries were evaluated – square, which supports modeling correlations between neighboring input regions, and mirror, which puts focus on correlations between regions that are symmetric w.r.t. one another. Each pooling geometry outperforms the other on the task for which its correlations are important, demonstrating how prior knowledge regarding a task at hand may be used to tailor the inductive bias through proper pooling design.

described above for the deep network besides the number of hidden channels, which in this case applied to the network's single hidden layer, and varied between 64 and 4096. The highest train and test accuracies delivered by this network (with 4096 hidden channels) were roughly 62% on closedness task, and 77% on symmetry task. The fact that these accuracies are inferior to those of the deep network, even when the latter's pooling geometry is not optimal for the task at hand, complies with our analysis in sec. 5. Namely, it complies with the observation that separation ranks (correlations) are sometimes exponential and sometimes polynomial with the deep network, whereas with the shallow one they are never more than linear in network size.

Finally, to assess the validity of our findings for convolutional networks in general, not just convolutional arithmetic circuits, we repeated the above experiments with convolutional rectifier networks. Namely, we placed ReLU activations after every conv operator, switched the pooling operation from product to average, and re-evaluated the deep (square and mirror pooling geometries) and shallow networks. We then reiterated this process once more, with pooling operation set to max instead of average. The results obtained by the deep networks are presented in fig. 4. The shallow network with average pooling reached train/test accuracies of roughly 58% on closedness task, and 55% on symmetry task. With max pooling, performance of the shallow network did not exceed chance. Altogether, convolutional rectifier networks exhibit the same phenomena observed with convolutional arithmetic circuits, indicating that the conclusions from our analyses likely apply to such networks as well. Formal adaptation of the analyses to convolutional rectifier networks, similarly to the adaptation of Cohen et al. (2016b) carried out in Cohen and Shashua (2016), is left for future work.

# 8 DISCUSSION

Through the notion of separation rank, we studied the relation between the architecture of a convolutional network, and its ability to model correlations among input regions. For a given input partition, the separation rank quantifies how far a function is from separability, which in a probabilistic setting, corresponds to statistical independence between sides of the partition.

Our analysis shows that a polynomially sized deep convolutional arithmetic circuit supports exponentially high separation ranks for certain input partitions, while being limited to polynomial or linear (in network size) separation ranks for others. The network's pooling window shapes effectively determine which input partitions are favored in terms of separation rank, *i.e.* which partitions enjoy the possibility of exponentially high separation ranks with polynomial network size, and which require network to be exponentially large. Pooling geometry thus serves as a means for controlling the inductive bias. The particular pooling scheme commonly employed in practice – square contiguous windows, favors interleaved partitions over ones that divide the input to distinct areas, thus orients the inductive bias towards the statistics of natural images (nearby pixels more correlated than distant

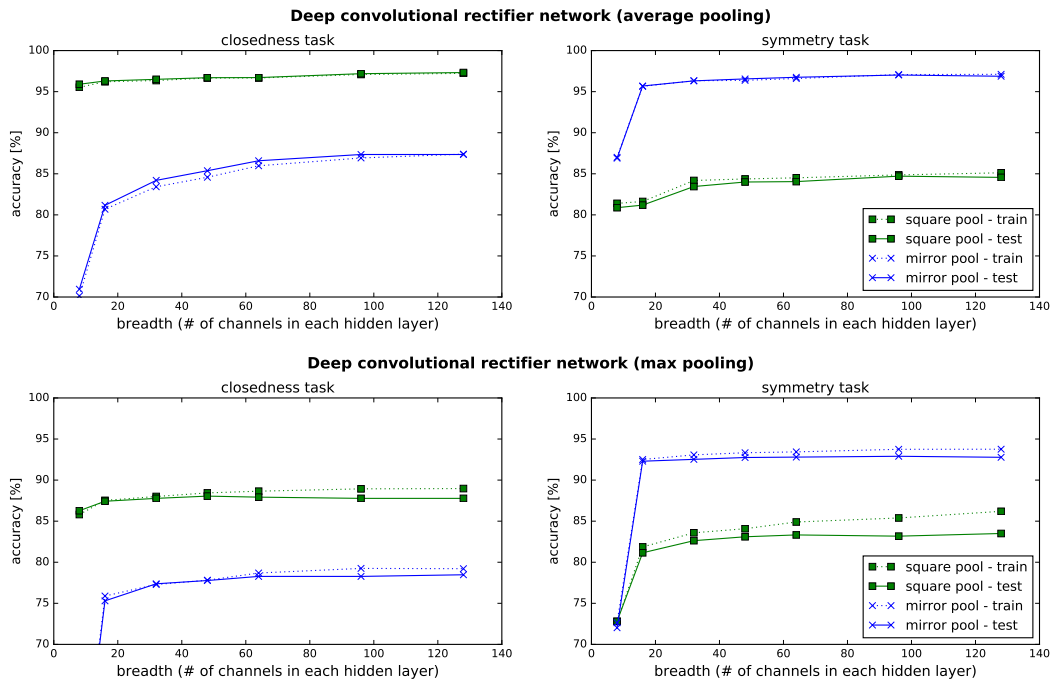

Figure 4: Results of applying deep convolutional rectifier networks to closedness and symmetry classification tasks. The same trends observed with the deep convolutional arithmetic circuit (fig. 3) are apparent here.

ones). Other pooling schemes lead to different preferences, and this allows tailoring the network to data that departs from the usual domain of natural imagery.

As opposed to deep convolutional arithmetic circuits, shallow ones support only linear (in network size) separation ranks. Therefore, in order to replicate a function realized by a deep network (exponential separation rank), a shallow network must be exponentially large. By this we derive the depth efficiency result of Cohen et al. (2016b), but in addition, provide an insight into the benefit of functions brought forth by depth – they are able to efficiently model strong correlation under favored partitions of the input.

We validated our conclusions empirically, with convolutional arithmetic circuits as well as convolutional rectifier networks – convolutional networks with ReLU activation and max or average pooling. Our experiments demonstrate how different pooling geometries lead to superior performance in different tasks. Specifically, we evaluate deep networks in the measurement of shape continuity, a task of a local nature, and show that standard square pooling windows outperform ones that join together nodes with their spatial reflections. In contrast, when measuring shape symmetry, modeling correlations across distances is of vital importance, and the latter pooling geometry is superior to the conventional one. Shallow networks are inefficient at modeling correlations of any kind, and indeed lead to poor performance on both tasks.

Finally, our analyses and results bring forth the possibility of expanding the coverage of correlations efficiently modeled by a deep convolutional network. Specifically, by blending together multiple pooling geometries in the hidden layers of a network, it is possible to facilitate simultaneous support for a wide variety of correlations suiting data of different types. Investigation of this direction, from both theoretical and empirical perspectives, is viewed as a promising avenue for future research.

ACKNOWLEDGMENTS

This work is supported by Intel grant ICRI-CI #9-2012-6133, by ISF Center grant 1790/12 and by the European Research Council (TheoryDL project). Nadav Cohen is supported by a Google Doctoral Fellowship in Machine Learning.

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

## A    DEFERRED PROOFS

### A.1    PROOF OF CLAIM 1

We prove the equality in two steps, first showing that $sep(h_y; I, J) \leq rank[\![\mathcal{A}^y]\!]_{I,J}$, and then establishing the converse. The first step is elementary, and does not make use of the representation functions' $(f_{\theta_d})$ linear independence, or of measurability/square-integrability. The second step does rely on these assumptions, and employs slightly more advanced mathematical machinery. Throughout the proof, we assume without loss of generality that the partition $(I, J)$ of $[N]$ is such that $I$ takes on lower values, while $J$ takes on higher ones. That is to say, we assume that $I = \{1, \ldots, |I|\}$ and $J = \{|I| + 1, \ldots, N\}$. [9]

To prove that $sep(h_y; I, J) \leq rank[\![\mathcal{A}^y]\!]_{I,J}$, denote by $R$ the rank of $[\![\mathcal{A}^y]\!]_{I,J}$. The latter is an $M^{|I|}$-by-$M^{|J|}$ matrix, thus there exist vectors $\mathbf{u}_1 \ldots \mathbf{u}_R \in \mathbb{R}^{M^{|I|}}$ and $\mathbf{v}_1 \ldots \mathbf{v}_R \in \mathbb{R}^{M^{|J|}}$ such that $[\![\mathcal{A}^y]\!]_{I,J} = \sum_{\nu=1}^R \mathbf{u}_\nu \mathbf{v}_\nu^\top$. For every $\nu \in [R]$, let $\mathcal{B}^\nu$ be the tensor of order $|I|$ and dimension $M$ in each mode whose arrangement as a column vector gives $\mathbf{u}_\nu$, *i.e.* whose matricization w.r.t. the partition $([|I|], \emptyset)$ is equal to $\mathbf{u}_\nu$. Similarly, let $\mathcal{C}^\nu$, $\nu \in [R]$, be the tensor of order $|J| = N - |I|$ and dimension $M$ in each mode whose matricization w.r.t. the partition $(\emptyset, [|J|])$ (arrangement as a row vector) is equal to $\mathbf{v}_\nu^\top$. It holds that:

$$
\begin{aligned}
[\![\mathcal{A}^y]\!]_{I,J} &= \sum_{\nu=1}^R \mathbf{u}_\nu \mathbf{v}_\nu^\top \\
&= \sum_{\nu=1}^R [\![\mathcal{B}^\nu]\!]_{[|I|],\emptyset} \odot [\![\mathcal{C}^\nu]\!]_{\emptyset,[|J|]} \\
&= \sum_{\nu=1}^R [\![\mathcal{B}^\nu]\!]_{I\cap[|I|],J\cap[|I|]} \odot [\![\mathcal{C}^\nu]\!]_{(I-|I|)\cap[|J|],(J-|I|)\cap[|J|]} \\
&= \sum_{\nu=1}^R [\![\mathcal{B}^\nu \otimes \mathcal{C}^\nu]\!]_{I,J} \\
&= \left[\!\!\left[ \sum_{\nu=1}^R \mathcal{B}^\nu \otimes \mathcal{C}^\nu \right]\!\!\right]_{I,J}
\end{aligned}
$$

where the third equality relies on the assumption $I = \{1, \ldots, |I|\}$, $J = \{|I| + 1, \ldots, N\}$, the fourth equality makes use of the relation in eq. 1, and the last equality is based on the linearity of the matricization operator. Since matricizations are merely rearrangements of tensors, the fact that $[\![\mathcal{A}^y]\!]_{I,J} = [\![\sum_{\nu=1}^R \mathcal{B}^\nu \otimes \mathcal{C}^\nu]\!]_{I,J}$ implies $\mathcal{A}^y = \sum_{\nu=1}^R \mathcal{B}^\nu \otimes \mathcal{C}^\nu$, or equivalently, $\mathcal{A}^y_{d_1 \ldots d_N} = \sum_{\nu=1}^R \mathcal{B}^\nu_{d_1 \ldots d_{|I|}} \cdot \mathcal{C}^\nu_{d_{|I|+1} \ldots d_N}$ for every $d_1 \ldots d_N \in [M]$. Plugging this into eq. 2 gives:

$$
\begin{aligned}
h_y(\mathbf{x}_1, \ldots, \mathbf{x}_N) &= \sum_{d_1 \ldots d_N = 1}^M \mathcal{A}^y_{d_1 \ldots d_N} \prod_{i=1}^N f_{\theta_{d_i}}(\mathbf{x}_i) \\
&= \sum_{d_1 \ldots d_N = 1}^M \sum_{\nu=1}^R \mathcal{B}^\nu_{d_1 \ldots d_{|I|}} \cdot \mathcal{C}^\nu_{d_{|I|+1} \ldots d_N} \prod_{i=1}^N f_{\theta_{d_i}}(\mathbf{x}_i) \\
&= \sum_{\nu=1}^R \left( \sum_{d_1 \ldots d_{|I|} = 1}^M \mathcal{B}^\nu_{d_1 \ldots d_{|I|}} \prod_{i=1}^{|I|} f_{\theta_{d_i}}(\mathbf{x}_i) \right) \\
&\qquad \cdot \left( \sum_{d_{|I|+1} \ldots d_N = 1}^M \mathcal{C}^\nu_{d_{|I|+1} \ldots d_N} \prod_{i=|I|+1}^N f_{\theta_{d_i}}(\mathbf{x}_i) \right) \quad (11)
\end{aligned}
$$

For every $\nu \in [R]$, define the functions $g_\nu : (\mathbb{R}^s)^{|I|} \to \mathbb{R}$ and $g'_\nu : (\mathbb{R}^s)^{|J|} \to \mathbb{R}$ as follows:

$$
\begin{aligned}
g_\nu(\mathbf{x}_1, \ldots, \mathbf{x}_{|I|}) &:= \sum_{d_1 \ldots d_{|I|} = 1}^M \mathcal{B}^\nu_{d_1 \ldots d_{|I|}} \prod_{i=1}^{|I|} f_{\theta_{d_i}}(\mathbf{x}_i) \\
g'_\nu(\mathbf{x}_1, \ldots, \mathbf{x}_{|J|}) &:= \sum_{d_1 \ldots d_{|J|} = 1}^M \mathcal{C}^\nu_{d_1 \ldots d_{|J|}} \prod_{i=1}^{|J|} f_{\theta_{d_i}}(\mathbf{x}_i)
\end{aligned}
$$

---

[9] To see that this does not limit generality, denote $I = \{i_1, \ldots, i_{|I|}\}$ and $J = \{j_1, \ldots, j_{|J|}\}$, and define an auxiliary function $h'_y$ by permuting the entries of $h_y$ such that those indexed by $I$ are on the left and those indexed by $J$ on the right, *i.e.* $h'_y(\mathbf{x}_{i_1}, \ldots, \mathbf{x}_{i_{|I|}}, \mathbf{x}_{j_1}, \ldots, \mathbf{x}_{j_{|J|}}) = h_y(\mathbf{x}_1, \ldots, \mathbf{x}_N)$. Obviously $sep(h_y; I, J) = sep(h'_y; I', J')$, where the partition $(I', J')$ is defined by $I' = \{1, \ldots, |I|\}$ and $J' = \{|I| + 1, \ldots, N\}$. Analogously to the definition of $h'_y$, let $\mathcal{A}'^y$ be the tensor obtained by permuting the modes of $\mathcal{A}^y$ such that those indexed by $I$ are on the left and those indexed by $J$ on the right, *i.e.* $\mathcal{A}'^y_{d_{i_1} \ldots d_{i_{|I|}} d_{j_1} \ldots d_{j_{|J|}}} = \mathcal{A}^y_{d_1 \ldots d_N}$. It is not difficult to see that matricizing $\mathcal{A}'^y$ w.r.t. $(I', J')$ is equivalent to matricizing $\mathcal{A}^y$ w.r.t. $(I, J)$, *i.e.* $[\![\mathcal{A}'^y]\!]_{I',J'} = [\![\mathcal{A}^y]\!]_{I,J}$, and in particular $rank[\![\mathcal{A}'^y]\!]_{I',J'} = rank[\![\mathcal{A}^y]\!]_{I,J}$. Moreover, since by definition $\mathcal{A}^y$ is a coefficient tensor corresponding to $h_y$ (eq. 2), $\mathcal{A}'^y$ will be a coefficient tensor that corresponds to $h'_y$. Now, our proof will show that $sep(h'_y; I', J') = rank[\![\mathcal{A}'^y]\!]_{I',J'}$, which, in light of the equalities above, implies $sep(h_y; I, J) = rank[\![\mathcal{A}^y]\!]_{I,J}$, as required.

Substituting these into eq. 11 leads to:

$$h_y\left(\mathbf{x}_1,\ldots,\mathbf{x}_N\right) = \sum\nolimits_{\nu=1}^{R} g_\nu(\mathbf{x}_1,\ldots,\mathbf{x}_{|I|})g'_\nu(\mathbf{x}_{|I|+1},\ldots,\mathbf{x}_N)$$

which by definition of the separation rank (eq. 5), implies $sep(h_y; I, J)\leq R$. By this we have shown that $sep(h_y; I, J)\leq rank[\![\mathcal{A}^y]\!]_{I,J}$, as required.

For proving the converse inequality, *i.e.* $sep(h_y; I, J)\geq rank[\![\mathcal{A}^y]\!]_{I,J}$, we rely on basic concepts and results from functional analysis, or more specifically, from the topic of $L^2$ spaces. While a full introduction to this topic is beyond our scope (the interested reader is referred to Rudin (1991)), we briefly lay out here the minimal background required in order to follow our proof. For any $n \in \mathbb{N}$, $L^2(\mathbb{R}^n)$ is formally defined as the Hilbert space of Lebesgue measurable square-integrable real functions over $\mathbb{R}^n$ [10] , equipped with standard (point-wise) addition and scalar multiplication, as well as the inner product defined by integration over point-wise multiplication. For our purposes, $L^2(\mathbb{R}^n)$ may simply be thought of as the (infinite-dimensional) vector space of functions $g : \mathbb{R}^n \to \mathbb{R}$ satisfying $\int g^2 < \infty$, with inner product defined by $\langle g_1, g_2 \rangle := \int g_1 \cdot g_2$. Our proof will make use of the following basic facts related to $L^2$ spaces:

**Fact 1.** *If $V$ is a finite-dimensional subspace of $L^2(\mathbb{R}^n)$, then any $g \in L^2(\mathbb{R}^n)$ may be expressed as $g = p + \delta$, with $p \in V$ and $\delta \in V^\perp$ (i.e. $\delta$ is orthogonal to all elements in $V$). Moreover, such a representation is unique, so in the case where $g \in V$, we necessarily have $p = g$ and $\delta \equiv 0$.*

**Fact 2.** *If $g \in L^2(\mathbb{R}^n)$, $g' \in L^2(\mathbb{R}^{n'})$, then the function $(\mathbf{x}_1, \mathbf{x}_2) \mapsto g(\mathbf{x}_1) \cdot g'(\mathbf{x}_2)$ belongs to $L^2(\mathbb{R}^n \times \mathbb{R}^{n'})$.*

**Fact 3.** *Let $V$ and $V'$ be finite-dimensional subspaces of $L^2(\mathbb{R}^n)$ and $L^2(\mathbb{R}^{n'})$ respectively, and define $U \subset L^2(\mathbb{R}^n \times \mathbb{R}^{n'})$ to be the subspace spanned by $\{(\mathbf{x}_1, \mathbf{x}_2) \mapsto p(\mathbf{x}_1) \cdot p'(\mathbf{x}_2) : p \in V, p' \in V'\}$. Given $g \in L^2(\mathbb{R}^n), g' \in L^2(\mathbb{R}^{n'})$, consider the function $(\mathbf{x}_1, \mathbf{x}_2) \mapsto g(\mathbf{x}_1) \cdot g'(\mathbf{x}_2)$ in $L^2(\mathbb{R}^n \times \mathbb{R}^{n'})$. This function belongs to $U^\perp$ if $g \in V^\perp$ or $g' \in V'^\perp$.*

**Fact 4.** *If $g_1 \ldots g_m \in L^2(\mathbb{R}^n)$ are linearly independent, then for any $k \in \mathbb{N}$, the set of functions $\{(\mathbf{x}_1, \ldots, \mathbf{x}_k) \mapsto \prod_{i=1}^{k} g_{d_i}(\mathbf{x}_i)\}_{d_1 \ldots d_k \in [m]}$ is linearly independent in $L^2((\mathbb{R}^n)^k)$.*

To facilitate application of the theory of $L^2$ spaces, we now make use of the assumption that the network's representation functions $f_{\theta_d}$, as well as the functions $g_\nu, g'_\nu$ in the definition of separation rank (eq. 5), are measurable and square-integrable. Taking into account the expression given in eq. 2 for $h_y$, as well as fact 2 above, one readily sees that $f_{\theta_1} \ldots f_{\theta_M} \in L^2(\mathbb{R}^s)$ implies $h_y \in L^2((\mathbb{R}^s)^N)$. The separation rank $sep(h_y; I, J)$ will be the minimal non-negative integer $R$ such that there exist $g_1 \ldots g_R \in L^2((\mathbb{R}^s)^{|I|})$ and $g'_1 \ldots g'_R \in L^2((\mathbb{R}^s)^{|J|})$ for which:

$$h_y(\mathbf{x}_1, \ldots, \mathbf{x}_N) = \sum\nolimits_{\nu=1}^{R} g_\nu(\mathbf{x}_1, \ldots, \mathbf{x}_{|I|})g'_\nu(\mathbf{x}_{|I|+1}, \ldots, \mathbf{x}_N) \qquad (12)$$

We would like to show that $sep(h_y; I, J) \geq rank[\![\mathcal{A}^y]\!]_{I,J}$. Our strategy for achieving this will be to start from eq. 12, and derive an expression for $[\![\mathcal{A}^y]\!]_{I,J}$ comprising a sum of $R$ rank-1 matrices. As an initial step along this path, define the following finite-dimensional subspaces:

$$V \quad := \quad span\left\{(\mathbf{x}_1, \ldots, \mathbf{x}_{|I|}) \mapsto \prod_{i=1}^{|I|} f_{\theta_{d_i}}(\mathbf{x}_i)\right\}_{d_1 \ldots d_{|I|} \in [M]} \quad \subset L^2\left((\mathbb{R}^s)^{|I|}\right) \qquad (13)$$

$$V' \quad := \quad span\left\{(\mathbf{x}_1, \ldots, \mathbf{x}_{|J|}) \mapsto \prod_{i=1}^{|J|} f_{\theta_{d_i}}(\mathbf{x}_i)\right\}_{d_1 \ldots d_{|J|} \in [M]} \quad \subset L^2\left((\mathbb{R}^s)^{|J|}\right) \qquad (14)$$

$$U \quad := \quad span\left\{(\mathbf{x}_1, \ldots, \mathbf{x}_N) \mapsto \prod_{i=1}^{N} f_{\theta_{d_i}}(\mathbf{x}_i)\right\}_{d_1 \ldots d_N \in [M]} \quad \subset L^2\left((\mathbb{R}^s)^N\right) \qquad (15)$$

Notice that $h_y \in U$ (eq. 2), and that $U$ is the span of products from $V$ and $V'$, *i.e.*:

$$U = span\left\{(\mathbf{x}_1, \ldots, \mathbf{x}_N) \mapsto p(\mathbf{x}_1, \ldots, \mathbf{x}_{|I|}) \cdot p'(\mathbf{x}_{|I|+1}, \ldots, \mathbf{x}_N) : p \in V, p' \in V'\right\} \qquad (16)$$

Returning to eq. 12, we apply fact 1 to obtain orthogonal decompositions of $g_1 \ldots g_R$ w.r.t. $V$, and of $g'_1 \ldots g'_R$ w.r.t. $V'$. This gives $p_1 \ldots p_R \in V, \delta_1 \ldots \delta_R \in V^\perp, p'_1 \ldots p'_R \in V'$ and $\delta'_1 \ldots \delta'_R \in V'^\perp$, such that $g_\nu = p_\nu + \delta_\nu$ and

---

[10] More precisely, elements of the space are equivalence classes of functions, where two functions are considered equivalent if the set in $\mathbb{R}^n$ on which they differ has measure zero.

$g'_\nu = p'_\nu + \delta'_\nu$ for every $\nu \in [R]$. Plug this into eq. 12:

$$
\begin{aligned}
h_y(\mathbf{x}_1,\ldots,\mathbf{x}_N) &= \sum_{\nu=1}^{R} g_\nu(\mathbf{x}_1,\ldots,\mathbf{x}_{|I|})\cdot g'_\nu(\mathbf{x}_{|I|+1},\ldots,\mathbf{x}_N) \\
&= \sum_{\nu=1}^{R} \big(p_\nu(\mathbf{x}_1,\ldots,\mathbf{x}_{|I|}) + \delta_\nu(\mathbf{x}_1,\ldots,\mathbf{x}_{|I|})\big) \\
&\qquad \cdot \big(p'_\nu(\mathbf{x}_{|I|+1},\ldots,\mathbf{x}_N) + \delta'_\nu(\mathbf{x}_{|I|+1},\ldots,\mathbf{x}_N)\big) \\
&= \sum_{\nu=1}^{R} p_\nu(\mathbf{x}_1,\ldots,\mathbf{x}_{|I|})\cdot p'_\nu(\mathbf{x}_{|I|+1},\ldots,\mathbf{x}_N) \\
&\quad + \sum_{\nu=1}^{R} p_\nu(\mathbf{x}_1,\ldots,\mathbf{x}_{|I|})\cdot \delta'_\nu(\mathbf{x}_{|I|+1},\ldots,\mathbf{x}_N) \\
&\quad + \sum_{\nu=1}^{R} \delta_\nu(\mathbf{x}_1,\ldots,\mathbf{x}_{|I|})\cdot p'_\nu(\mathbf{x}_{|I|+1},\ldots,\mathbf{x}_N) \\
&\quad + \sum_{\nu=1}^{R} \delta_\nu(\mathbf{x}_1,\ldots,\mathbf{x}_{|I|})\cdot \delta'_\nu(\mathbf{x}_{|I|+1},\ldots,\mathbf{x}_N)
\end{aligned}
$$

Given that $U$ is the span of products from $V$ and $V'$ (eq. 16), and that $p_\nu \in V, \delta_\nu \in V^\perp, p'_\nu \in V', \delta'_\nu \in V'^\perp$, one readily sees that the first term in the latter expression belongs to $U$, while, according to fact 3, the second, third and fourth terms are orthogonal to $U$. We thus obtained an orthogonal decomposition of $h_y$ w.r.t. $U$. Since $h_y$ is contained in $U$, the orthogonal component must vanish (fact 1), and we amount at:

$$
h_y(\mathbf{x}_1,\ldots,\mathbf{x}_N) = \sum_{\nu=1}^{R} p_\nu(\mathbf{x}_1,\ldots,\mathbf{x}_{|I|})\cdot p'_\nu(\mathbf{x}_{|I|+1},\ldots,\mathbf{x}_N) \tag{17}
$$

For every $\nu \in [R]$, let $\mathcal{B}^\nu$ and $\mathcal{C}^\nu$ be coefficient tensors of $p_\nu$ and $p'_\nu$ w.r.t. the functions that span $V$ and $V'$ (eq. 13 and 14), respectively. Put formally, $\mathcal{B}^\nu$ and $\mathcal{C}^\nu$ are tensors of orders $|I|$ and $|J|$ (respectively), with dimension $M$ in each mode, meeting:

$$
\begin{aligned}
p_\nu(\mathbf{x}_1,\ldots,\mathbf{x}_{|I|}) &= \sum_{d_1\ldots d_{|I|}=1}^{M} \mathcal{B}^\nu_{d_1\ldots d_{|I|}} \prod_{i=1}^{|I|} f_{\theta_{d_i}}(\mathbf{x}_i) \\
p'_\nu(\mathbf{x}_1,\ldots,\mathbf{x}_{|J|}) &= \sum_{d_1\ldots d_{|J|}=1}^{M} \mathcal{C}^\nu_{d_1\ldots d_{|J|}} \prod_{i=1}^{|J|} f_{\theta_{d_i}}(\mathbf{x}_i)
\end{aligned}
$$

Substitute into eq. 17:

$$
\begin{aligned}
h_y(\mathbf{x}_1,\ldots,\mathbf{x}_N) &= \sum_{\nu=1}^{R} \left(\sum_{d_1\ldots d_{|I|}=1}^{M} \mathcal{B}^\nu_{d_1\ldots d_{|I|}} \prod_{i=1}^{|I|} f_{\theta_{d_i}}(\mathbf{x}_i)\right) \\
&\qquad \cdot \left(\sum_{d_{|I|+1}\ldots d_N=1}^{M} \mathcal{C}^\nu_{d_{|I|+1}\ldots d_N} \prod_{i=|I|+1}^{N} f_{\theta_{d_i}}(\mathbf{x}_i)\right) \\
&= \sum_{\nu=1}^{R} \sum_{d_1\ldots d_N=1}^{M} \mathcal{B}^\nu_{d_1\ldots d_{|I|}} \cdot \mathcal{C}^\nu_{d_{|I|+1}\ldots d_N} \prod_{i=1}^{N} f_{\theta_{d_i}}(\mathbf{x}_i) \\
&= \sum_{d_1\ldots d_N=1}^{M} \left(\sum_{\nu=1}^{R} \mathcal{B}^\nu_{d_1\ldots d_{|I|}} \cdot \mathcal{C}^\nu_{d_{|I|+1}\ldots d_N}\right) \prod_{i=1}^{N} f_{\theta_{d_i}}(\mathbf{x}_i)
\end{aligned}
$$

Compare this expression for $h_y$ to that given in eq. 2:

$$
\sum_{d_1\ldots d_N=1}^{M} \left(\sum_{\nu=1}^{R} \mathcal{B}^\nu_{d_1\ldots d_{|I|}} \cdot \mathcal{C}^\nu_{d_{|I|+1}\ldots d_N}\right) \prod_{i=1}^{N} f_{\theta_{d_i}}(\mathbf{x}_i) = \sum_{d_1\ldots d_N=1}^{M} \mathcal{A}^y_{d_1\ldots d_N} \prod_{i=1}^{N} f_{\theta_{d_i}}(\mathbf{x}_i) \tag{18}
$$

At this point we utilize the given linear independence of $f_{\theta_1}\ldots f_{\theta_M} \in L^2(\mathbb{R}^s)$, from which it follows (fact 4) that the functions spanning $U$ (eq. 15) are linearly independent in $L^2((\mathbb{R}^s)^N)$. Both sides of eq. 18 are linear combinations of these functions, thus their coefficients must coincide:

$$
\mathcal{A}^y_{d_1\ldots d_N} = \sum_{\nu=1}^{R} \mathcal{B}^\nu_{d_1\ldots d_{|I|}} \cdot \mathcal{C}^\nu_{d_{|I|+1}\ldots d_N}, \forall d_1\ldots d_N \in [M] \iff \mathcal{A}^y = \sum_{\nu=1}^{R} \mathcal{B}^\nu \otimes \mathcal{C}^\nu
$$

Matricizing the tensor equation on the right w.r.t. $(I,J)$ gives:

$$
\begin{aligned}
[\![\mathcal{A}^y]\!]_{I,J} &= \left[\!\!\left[\sum_{\nu=1}^{R} \mathcal{B}^\nu \otimes \mathcal{C}^\nu\right]\!\!\right]_{I,J} \\
&= \sum_{\nu=1}^{R} [\![\mathcal{B}^\nu \otimes \mathcal{C}^\nu]\!]_{I,J} \\
&= \sum_{\nu=1}^{R} [\![\mathcal{B}^\nu]\!]_{I\cap[|I|],J\cap[|I|]} \odot [\![\mathcal{C}^\nu]\!]_{(I-|I|)\cap[|J|],(J-|I|)\cap[|J|]} \\
&= \sum_{\nu=1}^{R} [\![\mathcal{B}^\nu]\!]_{[|I|],\emptyset} \odot [\![\mathcal{C}^\nu]\!]_{\emptyset,[|J|]}
\end{aligned}
$$

where the second equality is based on the linearity of the matricization operator, the third equality relies on the relation in eq. 1, and the last equality makes use of the assumption $I = \{1,\ldots,|I|\}, J = \{|I|+1,\ldots,N\}$.

For every $\nu \in [R]$, $[\![\mathcal{B}^\nu]\!]_{[|I|],\emptyset}$ is a column vector of dimension $M^{|I|}$ and $[\![\mathcal{C}^\nu]\!]_{\emptyset,[|J|]}$ is a row vector of dimension $M^{|J|}$. Denoting these by $\mathbf{u}_\nu$ and $\mathbf{v}_\nu^\top$ respectively, we may write:

$$[\![\mathcal{A}^y]\!]_{I,J} = \sum\nolimits_{\nu=1}^{R} \mathbf{u}_\nu \mathbf{v}_\nu^\top$$

This shows that $rank[\![\mathcal{A}^y]\!]_{I,J} \leq R$. Since $R$ is a general non-negative integer that admits eq. 12, we may take it to be minimal, *i.e.* to be equal to $sep(h_y; I, J)$ – the separation rank of $h_y$ w.r.t. $(I, J)$. By this we obtain $rank[\![\mathcal{A}^y]\!]_{I,J} \leq sep(h_y; I, J)$, which is what we set out to prove.

$\square$

## A.2 PROOF OF CLAIM 2

The claim is framed in measure theoretical terms, and in accordance, so will its proof be. While a complete introduction to measure theory is beyond our scope (the interested reader is referred to Jones (2001)), we briefly convey here the intuition behind the concepts we will be using, as well as facts we rely upon. The *Lebesgue measure* is defined over sets in a Euclidean space, and may be interpreted as quantifying their "volume". For example, the Lebesgue measure of a unit hypercube is one, of the entire space is infinity, and of a finite set of points is zero. In this context, when a phenomenon is said to occur *almost everywhere*, it means that the set of points in which it does not occur has Lebesgue measure zero, *i.e.* is negligible. An important result we will make use of (proven in Caron and Traynor (2005) for example) is the following. Given a polynomial defined over $n$ real variables, the set of points in $\mathbb{R}^n$ on which it vanishes is either the entire space (when the polynomial in question is the zero polynomial), or it must have Lebesgue measure zero. In other words, if a polynomial is not identically zero, it must be different from zero almost everywhere.

Heading on to the proof, we recall from sec. 3 that the entries of the coefficient tensor $\mathcal{A}^y$ (eq. 2) are given by polynomials in the network's conv weights $\{\mathbf{a}^{l,\gamma}\}_{l,\gamma}$ and output weights $\mathbf{a}^{L,y}$. Since $[\![\mathcal{A}^y]\!]_{I,J}$ – the matricization of $\mathcal{A}^y$ w.r.t. the partition $(I, J)$, is merely a rearrangement of the tensor as a matrix, this matrix too has entries given by polynomials in the network's linear weights. Now, denote by $r$ the maximal rank taken by $[\![\mathcal{A}^y]\!]_{I,J}$ as network weights vary, and consider a specific setting of weights for which this rank is attained. We may assume without loss of generality that under this setting, the top-left $r$-by-$r$ block of $[\![\mathcal{A}^y]\!]_{I,J}$ is non-singular. The corresponding minor, *i.e.* the determinant of the sub-matrix $([\![\mathcal{A}^y]\!]_{I,J})_{1:r,1:r}$, is thus a polynomial defined over $\{\mathbf{a}^{l,\gamma}\}_{l,\gamma}$ and $\mathbf{a}^{L,y}$ which is not identically zero. In light of the above, this polynomial is different from zero almost everywhere, implying that $rank(([\![\mathcal{A}^y]\!]_{I,J})_{1:r,1:r}) = r$ almost everywhere. Since $rank[\![\mathcal{A}^y]\!]_{I,J} \geq rank(([\![\mathcal{A}^y]\!]_{I,J})_{1:r,1:r})$, and since by definition $r$ is the maximal rank that $[\![\mathcal{A}^y]\!]_{I,J}$ can take, we have that $rank[\![\mathcal{A}^y]\!]_{I,J}$ is maximal almost everywhere.

$\square$

## A.3 PROOF OF THEOREM 1

The matrix decomposition in eq. 7 expresses $[\![\mathcal{A}]\!]_{I,J}$ in terms of the network's linear weights – $\{\mathbf{a}^{0,\gamma} \in \mathbb{R}^M\}_{\gamma \in [r_0]}$ for conv operator in hidden layer 0, $\{\mathbf{a}^{l,\gamma} \in \mathbb{R}^{r_{l-1}}\}_{\gamma \in [r_l]}$ for conv operator in hidden layer $l = 1 \ldots L-1$, and $\mathbf{a}^{L,y} \in \mathbb{R}^{r_{L-1}}$ for node $y$ of dense output operator. We prove lower and upper bounds on the maximal rank that $[\![\mathcal{A}]\!]_{I,J}$ can take as these weights vary. Our proof relies on the rank-multiplicative property of the Kronecker product ($rank(A \odot B) = rank(A) \cdot rank(B)$ for any real matrices $A$ and $B$ – see Bellman (1970) for proof), but is otherwise elementary.

Beginning with the lower bound, consider the following weight setting ($\mathbf{e}_\gamma$ here stands for a vector holding 1 in entry $\gamma$ and 0 at all other entries, $\mathbf{0}$ stands for a vector holding 0 at all entries, and $\mathbf{1}$ stands for a vector holding 1 at all entries, with the dimension of a vector to be understood by context):

$$
\begin{aligned}
\mathbf{a}^{0,\gamma} &= \begin{cases} \mathbf{e}_\gamma & , \gamma \leq \min\{r_0, M\} \\ \mathbf{0} & , \text{otherwise} \end{cases} \\[4pt]
\mathbf{a}^{1,\gamma} &= \begin{cases} \mathbf{1} & , \gamma = 1 \\ \mathbf{0} & , \text{otherwise} \end{cases} \\[4pt]
\mathbf{a}^{l,\gamma} &= \begin{cases} \mathbf{e}_1 & , \gamma = 1 \\ \mathbf{0} & , \text{otherwise} \end{cases} \quad \text{for } l = 2 \ldots L-1 \\[4pt]
\mathbf{a}^{L,y} &= \mathbf{e}_1
\end{aligned}
\tag{19}
$$

Let $n \in [N/4]$. Recalling the definition of $I_{l,k}$ and $J_{l,k}$ from eq. 6, consider the sets $I_{1,n}$ and $J_{1,n}$, as well as $I_{0,4(n-1)+t}$ and $J_{0,4(n-1)+t}$ for $t \in [4]$. $(I_{1,n}, J_{1,n})$ is a partition of $[4]$, *i.e.* $I_{1,n} \cup J_{1,n} = [4]$, and for every $t \in [4]$ we have $I_{0,4(n-1)+t} = \{1\}$ and $J_{0,4(n-1)+t} = \emptyset$ if $t$ belongs to $I_{1,n}$, and otherwise $I_{0,4(n-1)+t} = \emptyset$

and $J_{0,4(n-1)+t} = \{1\}$ if $t$ belongs to $J_{1,n}$. This implies that for an arbitrary vector $\mathbf{v}$, the matricization $[\![\mathbf{v}]\!]_{I_{0,4(n-1)+t},J_{0,4(n-1)+t}}$ is equal to $\mathbf{v}$ if $t \in I_{1,n}$, and to $\mathbf{v}^\top$ if $t \in J_{1,n}$. Accordingly, for any $\gamma \in [r_0]$:

$$\overset{4}{\underset{t=1}{\odot}}[\![\mathbf{a}^{0,\gamma}]\!]_{I_{0,4(n-1)+t},J_{0,4(n-1)+t}} = \begin{cases} (\mathbf{a}^{0,\gamma} \odot \mathbf{a}^{0,\gamma} \odot \mathbf{a}^{0,\gamma} \odot \mathbf{a}^{0,\gamma}) & , |I_{1,n}| = 4 \; |J_{1,n}| = 0 \\ (\mathbf{a}^{0,\gamma} \odot \mathbf{a}^{0,\gamma} \odot \mathbf{a}^{0,\gamma})(\mathbf{a}^{0,\gamma})^\top & , |I_{1,n}| = 3 \; |J_{1,n}| = 1 \\ (\mathbf{a}^{0,\gamma} \odot \mathbf{a}^{0,\gamma})(\mathbf{a}^{0,\gamma} \odot \mathbf{a}^{0,\gamma})^\top & , |I_{1,n}| = 2 \; |J_{1,n}| = 2 \\ (\mathbf{a}^{0,\gamma})(\mathbf{a}^{0,\gamma} \odot \mathbf{a}^{0,\gamma} \odot \mathbf{a}^{0,\gamma})^\top & , |I_{1,n}| = 1 \; |J_{1,n}| = 3 \\ (\mathbf{a}^{0,\gamma} \odot \mathbf{a}^{0,\gamma} \odot \mathbf{a}^{0,\gamma} \odot \mathbf{a}^{0,\gamma})^\top & , |I_{1,n}| = 0 \; |J_{1,n}| = 4 \end{cases}$$

Assume that $\gamma \le \min\{r_0, M\}$. By our setting $\mathbf{a}^{0,\gamma} = \mathbf{e}_\gamma$, so the above matrix holds 1 in a single entry and 0 in all the rest. Moreover, if the matrix is not a row or column vector, *i.e.* if both $I_{1,n}$ and $J_{1,n}$ are non-empty, the column index and row index of the entry holding 1 are both unique w.r.t. $\gamma$, *i.e.* they do not repeat as $\gamma$ ranges over $1 \ldots \min\{r_0, M\}$. We thus have:

$$rank\left(\sum_{\gamma=1}^{\min\{r_0,M\}} \overset{4}{\underset{t=1}{\odot}}[\![\mathbf{a}^{0,\gamma}]\!]_{I_{0,4(n-1)+t},J_{0,4(n-1)+t}}\right) = \begin{cases} \min\{r_0, M\} & , I_{1,n} \ne \emptyset \wedge J_{1,n} \ne \emptyset \\ 1 & , I_{1,n} = \emptyset \vee J_{1,n} = \emptyset \end{cases}$$

Since we set $\mathbf{a}^{1,1} = \mathbf{1}$ and $\mathbf{a}^{0,\gamma} = \mathbf{0}$ for $\gamma > \min\{r_0, M\}$, we may write:

$$rank\left(\sum_{\gamma=1}^{r_0} a_\gamma^{1,1} \cdot \overset{4}{\underset{t=1}{\odot}}[\![\mathbf{a}^{0,\gamma}]\!]_{I_{0,4(n-1)+t},J_{0,4(n-1)+t}}\right) = \begin{cases} \min\{r_0, M\} & , I_{1,n} \ne \emptyset \wedge J_{1,n} \ne \emptyset \\ 1 & , I_{1,n} = \emptyset \vee J_{1,n} = \emptyset \end{cases}$$

The latter matrix is by definition equal to $[\![\phi^{1,1}]\!]_{I_{1,n},J_{1,n}}$ (see top row of eq. 7), and so for every $n \in [N/4]$:

$$rank\,[\![\phi^{1,1}]\!]_{I_{1,n},J_{1,n}} = \begin{cases} \min\{r_0, M\} & , I_{1,n} \ne \emptyset \wedge J_{1,n} \ne \emptyset \\ 1 & , I_{1,n} = \emptyset \vee J_{1,n} = \emptyset \end{cases} \tag{20}$$

Now, the fact that we set $\mathbf{a}^{L,y} = \mathbf{e}_1$ and $\mathbf{a}^{l,1} = \mathbf{e}_1$ for $l = 2 \ldots L-1$, implies that the second to last levels of the decomposition in eq. 7 collapse to:

$$[\![\mathcal{A}^y]\!]_{I,J} = \overset{N/4}{\underset{t=1}{\odot}}[\![\phi^{1,1}]\!]_{I_{1,t},J_{1,t}}$$

Applying the rank-multiplicative property of the Kronecker product, and plugging in eq. 20, we obtain:

$$rank[\![\mathcal{A}^y]\!]_{I,J} = \prod_{t=1}^{N/4} rank[\![\phi^{1,1}]\!]_{I_{1,t},J_{1,t}} = \min\{r_0, M\}^S$$

where $S := |\{t \in [N/4] : I_{1,t} \ne \emptyset \wedge J_{1,t} \ne \emptyset\}|$. This equality holds for the specific weight setting we defined in eq. 19. Maximizing over all weight settings gives the sought after lower bound:

$$\max_{\{\mathbf{a}^{l,\gamma}\}_{l,\gamma},\mathbf{a}^{L,y}} rank[\![\mathcal{A}^y]\!]_{I,J} \ge \min\{r_0, M\}^S$$

Moving on to the upper bound, we show by induction over $l = 1 \ldots L-1$ that for any $k \in [N/4^l]$ and $\gamma \in [r_l]$, the rank of $[\![\phi^{l,\gamma}]\!]_{I_{l,k},J_{l,k}}$ is no greater than $c^{l,k}$, regardless of the chosen weight setting. For the base case $l = 1$ we have:

$$[\![\phi^{1,\gamma}]\!]_{I_{1,k},J_{1,k}} = \sum_{\alpha=1}^{r_0} a_\alpha^{1,\gamma} \cdot \overset{4}{\underset{t=1}{\odot}}[\![\mathbf{a}^{0,\alpha}]\!]_{I_{0,4(k-1)+t},J_{0,4(k-1)+t}}$$

The $M^{|I_{1,k}|}$-by-$M^{|J_{1,k}|}$ matrix $[\![\phi^{1,\gamma}]\!]_{I_{1,k},J_{1,k}}$ is given here as a sum of $r_0$ rank-1 terms, thus obviously its rank is no greater than $\min\{M^{\min\{|I_{1,k}|,|J_{1,k}|\}}, r_0\}$. Since by definition $c^{0,t} = 1$ for all $t \in [N]$, we may write:

$$rank[\![\phi^{1,\gamma}]\!]_{I_{1,k},J_{1,k}} \le \min\left\{M^{\min\{|I_{1,k}|,|J_{1,k}|\}}, r_0 \prod_{t=1}^4 c^{0,4(k-1)+t}\right\}$$

$c^{1,k}$ is defined by the right hand side of this inequality, so our inductive hypotheses holds for $l = 1$. For $l > 1$:

$$[\![\phi^{l,\gamma}]\!]_{I_{l,k},J_{l,k}} = \sum_{\alpha=1}^{r_{l-1}} a_\alpha^{l,\gamma} \cdot \overset{4}{\underset{t=1}{\odot}}[\![\phi^{l-1,\alpha}]\!]_{I_{l-1,4(k-1)+t},J_{l-1,4(k-1)+t}}$$

Taking ranks:

$$\begin{aligned} rank[\![\phi^{l,\gamma}]\!]_{I_{l,k},J_{l,k}} &= rank\left(\sum_{\alpha=1}^{r_{l-1}} a_\alpha^{l,\gamma} \cdot \overset{4}{\underset{t=1}{\odot}}[\![\phi^{l-1,\alpha}]\!]_{I_{l-1,4(k-1)+t},J_{l-1,4(k-1)+t}}\right) \\ &\le \sum_{\alpha=1}^{r_{l-1}} rank\left(\overset{4}{\underset{t=1}{\odot}}[\![\phi^{l-1,\alpha}]\!]_{I_{l-1,4(k-1)+t},J_{l-1,4(k-1)+t}}\right) \\ &= \sum_{\alpha=1}^{r_{l-1}} \prod_{t=1}^4 rank[\![\phi^{l-1,\alpha}]\!]_{I_{l-1,4(k-1)+t},J_{l-1,4(k-1)+t}} \\ &\le \sum_{\alpha=1}^{r_{l-1}} \prod_{t=1}^4 c^{l-1,4(k-1)+t} \\ &= r_{l-1} \prod_{t=1}^4 c^{l-1,4(k-1)+t} \end{aligned}$$

where we used rank sub-additivity in the second line, the rank-multiplicative property of the Kronecker product in the third line, and our inductive hypotheses for $l-1$ in the fourth line. Since the number rows and columns in $[\![\phi^{l,\gamma}]\!]_{I_{l,k},J_{l,k}}$ is $M^{|I_{l,k}|}$ and $M^{|J_{l,k}|}$ respectively, we may incorporate these terms into the inequality, obtaining:

$$rank[\![\phi^{l,\gamma}]\!]_{I_{l,k},J_{l,k}} \leq \min\left\{ M^{\min\{|I_{l,k}|,|J_{l,k}|\}}, r_{l-1}\prod_{t=1}^{4} c^{l-1,4(k-1)+t}\right\}$$

The right hand side here is equal to $c^{l,k}$ by definition, so our inductive hypotheses indeed holds for all $l = 1\ldots L-1$. To establish the sought after upper bound on the rank of $[\![\mathcal{A}^y]\!]_{I,J}$, we recall that the latter is given by:

$$[\![\mathcal{A}^y]\!]_{I,J} = \sum_{\alpha=1}^{r_{L-1}} a_\alpha^{L,y} \cdot \overset{4}{\underset{t=1}{\odot}} [\![\phi^{L-1,\alpha}]\!]_{I_{L-1,t},J_{L-1,t}}$$

Carry out a series of steps similar to before, while making use of our inductive hypotheses for $l = L-1$:

$$
\begin{aligned}
rank[\![\mathcal{A}^y]\!]_{I,J} &= rank\left(\sum_{\alpha=1}^{r_{L-1}} a_\alpha^{L,y} \cdot \overset{4}{\underset{t=1}{\odot}} [\![\phi^{L-1,\alpha}]\!]_{I_{L-1,t},J_{L-1,t}}\right)\\
&\leq \sum_{\alpha=1}^{r_{L-1}} rank\left(\overset{4}{\underset{t=1}{\odot}} [\![\phi^{L-1,\alpha}]\!]_{I_{L-1,t},J_{L-1,t}}\right)\\
&= \sum_{\alpha=1}^{r_{L-1}} \prod_{t=1}^{4} rank[\![\phi^{L-1,\alpha}]\!]_{I_{L-1,t},J_{L-1,t}}\\
&\leq \sum_{\alpha=1}^{r_{L-1}} \prod_{t=1}^{4} c^{L-1,t}\\
&= r_{L-1}\prod_{t=1}^{4} c^{L-1,t}
\end{aligned}
$$

Since $[\![\mathcal{A}^y]\!]_{I,J}$ has $M^{|I|}$ rows and $M^{|J|}$ columns, we may include these terms in the inequality, thus reaching the upper bound we set out to prove.

$\square$

# B  Separation rank and the $L^2$ distance from separable functions

Our analysis of correlations modeled by convolutional networks is based on the concept of separation rank, conveyed in sec. 4. When the separation rank of a function w.r.t. a partition of its input is equal to 1, the function is separable, meaning it does not model any interaction between sides of the partition. We argued that the higher the separation rank, the farther the function is from this situation, *i.e.* the stronger the correlation it induces between sides of the partition. In the current appendix we formalize this argument, by relating separation rank to the $L^2$ distance from the set of separable functions. We begin by defining and characterizing a normalized (scale invariant) version of this distance (app. B.1). It is then shown (app. B.2) that separation rank provides an upper bound on the normalized distance. Finally, a lower bound that applies to deep convolutional arithmetic circuits is derived (app. B.3), based on the lower bound for their separation ranks established in sec. 5.2. Together, these steps imply that our entire analysis, facilitated by upper and lower bounds on separation ranks of convolutional arithmetic circuits, can be interpreted as based on upper and lower bounds on (normalized) $L^2$ distances from separable functions.

In the text hereafter, we assume familiarity of the reader with the contents of sec. 2, 3, 4, 5 and the proofs given in app. A. We also rely on basic knowledge in the topic of $L^2$ spaces (see discussion in app. A.1 for minimal background required in order to follow our arguments), as well as several results concerning singular values of matrices. In line with sec. 5, an assumption throughout this appendix is that all functions in question are measurable and square-integrable (*i.e.* belong to $L^2$ over the respective Euclidean space), and in app. B.3, we also make use of the fact that representation functions ($f_{\theta_d}$) of a convolutional arithmetic circuit can be regarded as linearly independent (see sec. 5.1). Finally, for convenience, we now fix $(I, J)$ – an arbitrary partition of $[N]$. Specifically, $I$ and $J$ are disjoint subsets of $[N]$ whose union gives $[N]$, denoted by $I = \{i_1, \ldots, i_{|I|}\}$ with $i_1 < \cdots < i_{|I|}$, and $J = \{j_1, \ldots, j_{|J|}\}$ with $j_1 < \cdots < j_{|J|}$.

## B.1  Normalized $L^2$ distance from separable functions

For a function $h \in L^2((\mathbb{R}^s)^N)$ (which is not identically zero), the *normalized $L^2$ distance from the set of separable functions w.r.t.* $(I, J)$, is defined as follows:

$$D(h; I, J) := \frac{1}{\|h\|} \cdot \inf_{\substack{g\in L^2((\mathbb{R}^s)^{|I|})\\ g'\in L^2((\mathbb{R}^s)^{|J|})}} \left\| h(\mathbf{x}_1, \ldots, \mathbf{x}_N) - g(\mathbf{x}_{i_1}, \ldots, \mathbf{x}_{i_{|I|}})g'(\mathbf{x}_{j_1}, \ldots, \mathbf{x}_{j_{|J|}})\right\| \quad (21)$$

where $\|\cdot\|$ refers to the norm of $L^2$ space, *e.g.* $\|h\| := (\int_{(\mathbb{R}^s)^N} h^2)^{1/2}$. In words, $D(h; I, J)$ is defined as the minimal $L^2$ distance between $h$ and a function that is separable w.r.t. $(I, J)$, divided by the norm of $h$. The

normalization (division by $\|h\|$) admits scale invariance to $D(h; I, J)$, and is of critical importance – without it, rescaling $h$ would accordingly rescale the distance measure, rendering the latter uninformative in terms of deviation from separability.

It is worthwhile noting the resemblance between $D(h; I, J)$ and the concept of mutual information (see Cover and Thomas (2012) for a comprehensive introduction). Both measures quantify the interaction that a normalized function [11] induces between input variables, by measuring distance from separable functions. The difference between the measures is threefold. First, mutual information considers probability density functions (non-negative and in $L^1$), while $D(h; I, J)$ applies to functions in $L^2$. Second, the notion of distance in mutual information is quantified through the Kullback-Leibler divergence, whereas in $D(h; I, J)$ it is simply the $L^2$ metric. Third, while mutual information evaluates the distance from a specific separable function – product of marginal distributions, $D(h; I, J)$ evaluates the minimal distance across all separable functions.

We now turn to establish a spectral characterization of $D(h; I, J)$, which will be used in app. B.2 and B.3 for deriving upper and lower bounds (respectively). Assume we have the following expression for $h$:

$$h(\mathbf{x}_1, \ldots, \mathbf{x}_N) = \sum_{\mu=1}^{m} \sum_{\mu'=1}^{m'} A_{\mu,\mu'} \cdot \phi_\mu(\mathbf{x}_{i_1}, \ldots, \mathbf{x}_{i_{|I|}}) \phi'_{\mu'}(\mathbf{x}_{j_1}, \ldots, \mathbf{x}_{j_{|J|}}) \tag{22}$$

where $m$ and $m'$ are positive integers, $A$ is an $m$-by-$m'$ real matrix, and $\{\phi_\mu\}_{\mu=1}^{m}$, $\{\phi'_{\mu'}\}_{\mu'=1}^{m'}$ are orthonormal sets of functions in $L^2((\mathbb{R}^s)^{|I|}), L^2((\mathbb{R}^s)^{|J|})$ respectively. We refer to such expression as an *orthonormal separable decomposition* of $h$, with $A$ being its *coefficient matrix*. We will show that for any orthonormal separable decomposition, $D(h; I, J)$ is given by the following formula:

$$D(h; I, J) = \sqrt{1 - \frac{\sigma_1^2(A)}{\sigma_1^2(A) + \cdots + \sigma_{\min\{m,m'\}}^2(A)}} \tag{23}$$

where $\sigma_1(A) \geq \cdots \geq \sigma_{\min\{m,m'\}}(A) \geq 0$ are the singular values of the coefficient matrix $A$. This implies that if the largest singular value of $A$ accounts for a significant portion of the spectral energy, the normalized $L^2$ distance of $h$ from separable functions is small. On the other hand, if all but a fraction of the spectral energy is attributed to trailing singular values, $h$ is far from being separable ($D(h; I, J)$ is close to 1).

As a first step in deriving eq. 23, we show that $\|h\|^2 = \sigma_1^2(A) + \cdots + \sigma_{\min\{m,m'\}}^2(A)$:

$$
\begin{aligned}
\|h\|^2 &\underset{(1)}{=} \int h^2(\mathbf{x}_1, \ldots, \mathbf{x}_N) d\mathbf{x}_1 \cdots d\mathbf{x}_N \\
&\underset{(2)}{=} \int \left( \sum_{\mu=1}^{m} \sum_{\mu'=1}^{m'} A_{\mu,\mu'} \cdot \phi_\mu(\mathbf{x}_{i_1}, \ldots, \mathbf{x}_{i_{|I|}}) \phi'_{\mu'}(\mathbf{x}_{j_1}, \ldots, \mathbf{x}_{j_{|J|}}) \right)^2 d\mathbf{x}_1 \cdots d\mathbf{x}_N \\
&\underset{(3)}{=} \int \sum_{\mu,\bar{\mu}=1}^{m} \sum_{\mu',\bar{\mu}'=1}^{m'} A_{\mu,\mu'} A_{\bar{\mu},\bar{\mu}'} \cdot \phi_\mu(\mathbf{x}_{i_1}, \ldots, \mathbf{x}_{i_{|I|}}) \phi'_{\mu'}(\mathbf{x}_{j_1}, \ldots, \mathbf{x}_{j_{|J|}}) \\
&\qquad\qquad\qquad\qquad\qquad \cdot \phi_{\bar{\mu}}(\mathbf{x}_{i_1}, \ldots, \mathbf{x}_{i_{|I|}}) \phi'_{\bar{\mu}'}(\mathbf{x}_{j_1}, \ldots, \mathbf{x}_{j_{|J|}}) d\mathbf{x}_1 \cdots d\mathbf{x}_N \\
&\underset{(4)}{=} \sum_{\mu,\bar{\mu}=1}^{m} \sum_{\mu',\bar{\mu}'=1}^{m'} A_{\mu,\mu'} A_{\bar{\mu},\bar{\mu}'} \int \phi_\mu(\mathbf{x}_{i_1}, \ldots, \mathbf{x}_{i_{|I|}}) \phi'_{\mu'}(\mathbf{x}_{j_1}, \ldots, \mathbf{x}_{j_{|J|}}) \\
&\qquad\qquad\qquad\qquad\qquad \cdot \phi_{\bar{\mu}}(\mathbf{x}_{i_1}, \ldots, \mathbf{x}_{i_{|I|}}) \phi'_{\bar{\mu}'}(\mathbf{x}_{j_1}, \ldots, \mathbf{x}_{j_{|J|}}) d\mathbf{x}_1 \cdots d\mathbf{x}_N \\
&\underset{(5)}{=} \sum_{\mu,\bar{\mu}=1}^{m} \sum_{\mu',\bar{\mu}'=1}^{m'} A_{\mu,\mu'} A_{\bar{\mu},\bar{\mu}'} \int \phi_\mu(\mathbf{x}_{i_1}, \ldots, \mathbf{x}_{i_{|I|}}) \phi_{\bar{\mu}}(\mathbf{x}_{i_1}, \ldots, \mathbf{x}_{i_{|I|}}) d\mathbf{x}_{i_1} \cdots d\mathbf{x}_{i_{|I|}} \\
&\qquad\qquad\qquad\qquad\qquad \cdot \int \phi'_{\mu'}(\mathbf{x}_{j_1}, \ldots, \mathbf{x}_{j_{|J|}}) \phi'_{\bar{\mu}'}(\mathbf{x}_{j_1}, \ldots, \mathbf{x}_{j_{|J|}}) d\mathbf{x}_{j_1} \cdots d\mathbf{x}_{j_{|J|}} \\
&\underset{(6)}{=} \sum_{\mu,\bar{\mu}=1}^{m} \sum_{\mu',\bar{\mu}'=1}^{m'} A_{\mu,\mu'} A_{\bar{\mu},\bar{\mu}'} \cdot \left\{ \begin{array}{ll} 1 & , \mu = \bar{\mu} \\ 0 & , \text{otherwise} \end{array} \right\} \cdot \left\{ \begin{array}{ll} 1 & , \mu' = \bar{\mu}' \\ 0 & , \text{otherwise} \end{array} \right\} \\
&\underset{(7)}{=} \sum_{\mu=1}^{m} \sum_{\mu'=1}^{m'} A_{\mu,\mu'}^2 \\
&\underset{(8)}{=} \sigma_1^2(A) + \cdots + \sigma_{\min\{m,m'\}}^2(A) \tag{24}
\end{aligned}
$$

Equality (1) here originates from the definition of $L^2$ norm. (2) is obtained by plugging in the expression in eq. 22. (3) is merely an arithmetic manipulation. (4) follows from the linearity of integration. (5) makes use

---

[11] An equivalent definition of $D(h; I, J)$ is the minimal $L^2$ distance between $h/\|h\|$ and a function separable w.r.t. $(I, J)$. Accordingly, we may view $D(h; I, J)$ as operating on normalized functions.

of Fubini's theorem (see Jones (2001)). (6) results from the orthonormality of $\{\phi_\mu\}_{\mu=1}^m$ and $\{\phi'_{\mu'}\}_{\mu'=1}^{m'}$. (7) is a trivial computation. Finally, (8) is an outcome of the fact that the squared Frobenius norm of a matrix, *i.e.* the sum of squares over its entries, is equal to the sum of squares over its singular values (see Golub and Van Loan (2013) for proof).

Let $g \in L^2((\mathbb{R}^s)^{|I|})$. By fact 1 in app. A.1, there exist scalars $\alpha_1 \ldots \alpha_m \in \mathbb{R}$, and a function $\delta \in L^2((\mathbb{R}^s)^{|I|})$ orthogonal to $span\{\phi_1 \ldots \phi_m\}$, such that $g = \sum_{\mu=1}^m \alpha_\mu \cdot \phi_\mu + \delta$. Similarly, for any $g' \in L^2((\mathbb{R}^s)^{|J|})$ there exist $\alpha'_1 \ldots \alpha'_{m'} \in \mathbb{R}$ and $\delta' \in span\{\phi'_1 \ldots \phi'_{m'}\}^\perp$ such that $g' = \sum_{\mu'=1}^{m'} \alpha'_{\mu'} \cdot \phi'_{\mu'} + \delta'$. Fact 2 in app. A.1 indicates that the function given by $(\mathbf{x}_1, \ldots, \mathbf{x}_N) \mapsto g(\mathbf{x}_{i_1}, \ldots, \mathbf{x}_{i_{|I|}})g'(\mathbf{x}_{j_1}, \ldots, \mathbf{x}_{j_{|J|}})$ belongs to $L^2((\mathbb{R}^s)^N)$. We may express it as follows:

$$
\begin{aligned}
g(\mathbf{x}_{i_1}, \ldots, \mathbf{x}_{i_{|I|}})g'(\mathbf{x}_{j_1}, \ldots, \mathbf{x}_{j_{|J|}}) = &\sum_{\mu=1}^m \sum_{\mu'=1}^{m'} \alpha_\mu \alpha'_{\mu'} \cdot \phi_\mu(\mathbf{x}_{i_1}, \ldots, \mathbf{x}_{i_{|I|}})\phi'_{\mu'}(\mathbf{x}_{j_1}, \ldots, \mathbf{x}_{j_{|J|}}) \\
&+ \left(\sum_{\mu=1}^m \alpha_\mu \cdot \phi_\mu(\mathbf{x}_{i_1}, \ldots, \mathbf{x}_{i_{|I|}})\right) \cdot \delta'(\mathbf{x}_{j_1}, \ldots, \mathbf{x}_{j_{|J|}}) \\
&+ \delta(\mathbf{x}_{i_1}, \ldots, \mathbf{x}_{i_{|I|}}) \cdot \left(\sum_{\mu'=1}^{m'} \alpha'_{\mu'} \cdot \phi'_{\mu'}(\mathbf{x}_{j_1}, \ldots, \mathbf{x}_{j_{|J|}})\right) \\
&+ \delta(\mathbf{x}_{i_1}, \ldots, \mathbf{x}_{i_{|I|}})\delta'(\mathbf{x}_{j_1}, \ldots, \mathbf{x}_{j_{|J|}})
\end{aligned}
$$

According to fact 3 in app. A.1, the second, third and fourth terms on the right hand side of the above are orthogonal to $span\{(\mathbf{x}_1, \ldots, \mathbf{x}_N) \mapsto \phi_\mu(\mathbf{x}_{i_1}, \ldots, \mathbf{x}_{i_{|I|}})\phi'_{\mu'}(\mathbf{x}_{j_1}, \ldots, \mathbf{x}_{j_{|J|}})\}_{\mu \in [m], \mu' \in [m']}$. Denote their summation by $\mathcal{E}(\mathbf{x}_1, \ldots, \mathbf{x}_N)$, and subtract the overall function from $h$ (given by eq. 22):

$$
\begin{aligned}
&h(\mathbf{x}_1, \ldots, \mathbf{x}_N) - g(\mathbf{x}_{i_1}, \ldots, \mathbf{x}_{i_{|I|}})g'(\mathbf{x}_{j_1}, \ldots, \mathbf{x}_{j_{|J|}}) \\
&= \sum_{\mu=1}^m \sum_{\mu'=1}^{m'} A_{\mu,\mu'} \cdot \phi_\mu(\mathbf{x}_{i_1}, \ldots, \mathbf{x}_{i_{|I|}})\phi'_{\mu'}(\mathbf{x}_{j_1}, \ldots, \mathbf{x}_{j_{|J|}}) \\
&\quad - \sum_{\mu=1}^m \sum_{\mu'=1}^{m'} \alpha_\mu \alpha'_{\mu'} \cdot \phi_\mu(\mathbf{x}_{i_1}, \ldots, \mathbf{x}_{i_{|I|}})\phi'_{\mu'}(\mathbf{x}_{j_1}, \ldots, \mathbf{x}_{j_{|J|}}) - \mathcal{E}(\mathbf{x}_1, \ldots, \mathbf{x}_N) \\
&= \sum_{\mu=1}^m \sum_{\mu'=1}^{m'} (A_{\mu,\mu'} - \alpha_\mu \alpha'_{\mu'}) \cdot \phi_\mu(\mathbf{x}_{i_1}, \ldots, \mathbf{x}_{i_{|I|}})\phi'_{\mu'}(\mathbf{x}_{j_1}, \ldots, \mathbf{x}_{j_{|J|}}) - \mathcal{E}(\mathbf{x}_1, \ldots, \mathbf{x}_N)
\end{aligned}
$$

Since the two terms in the latter expression are orthogonal to one another, we have:

$$
\begin{aligned}
&\left\|h(\mathbf{x}_1, \ldots, \mathbf{x}_N) - g(\mathbf{x}_{i_1}, \ldots, \mathbf{x}_{i_{|I|}})g'(\mathbf{x}_{j_1}, \ldots, \mathbf{x}_{j_{|J|}})\right\|^2 \\
&= \left\|\sum_{\mu=1}^m \sum_{\mu'=1}^{m'} (A_{\mu,\mu'} - \alpha_\mu \alpha'_{\mu'}) \cdot \phi_\mu(\mathbf{x}_{i_1}, \ldots, \mathbf{x}_{i_{|I|}})\phi'_{\mu'}(\mathbf{x}_{j_1}, \ldots, \mathbf{x}_{j_{|J|}})\right\|^2 + \|\mathcal{E}(\mathbf{x}_1, \ldots, \mathbf{x}_N)\|^2
\end{aligned}
$$

Applying a sequence of steps as in eq. 24 to the first term in the second line of the above, we obtain:

$$
\left\|h(\mathbf{x}_1, \ldots, \mathbf{x}_N) - g(\mathbf{x}_{i_1}, \ldots, \mathbf{x}_{i_{|I|}})g'(\mathbf{x}_{j_1}, \ldots, \mathbf{x}_{j_{|J|}})\right\|^2 = \sum_{\mu=1}^m \sum_{\mu'=1}^{m'} (A_{\mu,\mu'} - \alpha_\mu \alpha'_{\mu'})^2 + \|\mathcal{E}(\mathbf{x}_1, \ldots, \mathbf{x}_N)\|^2
$$

$\mathcal{E}(\mathbf{x}_1, \ldots, \mathbf{x}_N) = 0$ if $\delta$ and $\delta'$ are the zero functions, implying that:

$$
\left\|h(\mathbf{x}_1, \ldots, \mathbf{x}_N) - g(\mathbf{x}_{i_1}, \ldots, \mathbf{x}_{i_{|I|}})g'(\mathbf{x}_{j_1}, \ldots, \mathbf{x}_{j_{|J|}})\right\|^2 \geq \sum_{\mu=1}^m \sum_{\mu'=1}^{m'} (A_{\mu,\mu'} - \alpha_\mu \alpha'_{\mu'})^2
$$

with equality holding if $g = \sum_{\mu=1}^m \alpha_\mu \cdot \phi_\mu$ and $g' = \sum_{\mu'=1}^{m'} \alpha'_{\mu'} \cdot \phi'_{\mu'}$. Now, $\sum_{\mu=1}^m \sum_{\mu'=1}^{m'} (A_{\mu,\mu'} - \alpha_\mu \alpha'_{\mu'})^2$ is the squared Frobenius distance between the matrix $A$ and the rank-1 matrix $\boldsymbol{\alpha}\boldsymbol{\alpha}'^\top$, where $\boldsymbol{\alpha}$ and $\boldsymbol{\alpha}'$ are column vectors holding $\alpha_1 \ldots \alpha_m$ and $\alpha'_1 \ldots \alpha'_{m'}$ respectively. This squared distance is greater than or equal to the sum of squares over the second to last singular values of $A$, and moreover, the inequality holds with equality for proper choices of $\boldsymbol{\alpha}$ and $\boldsymbol{\alpha}'$ (Eckart and Young (1936)). From this we conclude that:

$$
\left\|h(\mathbf{x}_1, \ldots, \mathbf{x}_N) - g(\mathbf{x}_{i_1}, \ldots, \mathbf{x}_{i_{|I|}})g'(\mathbf{x}_{j_1}, \ldots, \mathbf{x}_{j_{|J|}})\right\|^2 \geq \sigma_2^2(A) + \cdots + \sigma_{\min\{m,m'\}}^2(A)
$$

with equality holding if $g$ and $g'$ are set to $\sum_{\mu=1}^m \alpha_\mu \cdot \phi_\mu$ and $\sum_{\mu'=1}^{m'} \alpha'_{\mu'} \cdot \phi'_{\mu'}$ (respectively) for proper choices of $\alpha_1 \ldots \alpha_m$ and $\alpha'_1 \ldots \alpha'_{m'}$. We thus have the infimum over all possible $g, g'$:

$$
\inf_{\substack{g \in L^2((\mathbb{R}^s)^{|I|}) \\ g' \in L^2((\mathbb{R}^s)^{|J|})}} \left\|h(\mathbf{x}_1, \ldots, \mathbf{x}_N) - g(\mathbf{x}_{i_1}, \ldots, \mathbf{x}_{i_{|I|}})g'(\mathbf{x}_{j_1}, \ldots, \mathbf{x}_{j_{|J|}})\right\|^2 = \sigma_2^2(A) + \cdots + \sigma_{\min\{m,m'\}}^2(A)
$$
(25)

Recall that we would like to derive the formula in eq. 23 for $D(h; I, J)$, assuming $h$ is given by the orthonormal separable decomposition in eq. 22. Taking square root of the equalities established in eq. 24 and 25, and plugging them into the definition of $D(h; I, J)$ (eq. 21), we obtain the sought after result.

## B.2 UPPER BOUND THROUGH SEPARATION RANK

We now relate $D(h; I, J)$ – the normalized $L^2$ distance of $h \in L^2((\mathbb{R}^s)^N)$ from the set of separable functions w.r.t. $(I, J)$ (eq. 21), to $sep(h; I, J)$ – the separation rank of $h$ w.r.t. $(I, J)$ (eq. 5). Specifically, we make use of the formula in eq. 23 to derive an upper bound on $D(h; I, J)$ in terms of $sep(h; I, J)$.

Assuming $h$ has finite separation rank (otherwise the bound we derive is trivial), we may express it as:

$$h(\mathbf{x}_1, \ldots, \mathbf{x}_N) = \sum\nolimits_{\nu=1}^{R} g_\nu(\mathbf{x}_{i_1}, \ldots, \mathbf{x}_{i_{|I|}}) g'_\nu(\mathbf{x}_{j_1}, \ldots, \mathbf{x}_{j_{|J|}}) \qquad (26)$$

where $R$ is some positive integer (necessarily greater than or equal to $sep(h; I, J)$), and $g_1 \ldots g_R \in L^2((\mathbb{R}^s)^{|I|})$, $g'_1 \ldots g'_R \in L^2((\mathbb{R}^s)^{|J|})$. Let $\{\phi_1, \ldots, \phi_m\} \subset L^2((\mathbb{R}^s)^{|I|})$ and $\{\phi'_1, \ldots, \phi'_{m'}\} \subset L^2((\mathbb{R}^s)^{|J|})$ be two sets of orthonormal functions spanning $span\{g_1 \ldots g_R\}$ and $span\{g'_1 \ldots g'_R\}$ respectively. By definition, for every $\nu \in R$ there exist $\alpha_{\nu,1} \ldots \alpha_{\nu,m} \in \mathbb{R}$ and $\alpha'_{\nu,1} \ldots \alpha'_{\nu,m'} \in \mathbb{R}$ such that $g_\nu = \sum_{\mu=1}^{m} \alpha_{\nu,\mu} \cdot \phi_\mu$ and $g'_\nu = \sum_{\mu'=1}^{m'} \alpha'_{\nu,\mu'} \cdot \phi'_{\mu'}$. Plugging this into eq. 26, we obtain:

$$h(\mathbf{x}_1, \ldots, \mathbf{x}_N) = \sum\nolimits_{\mu=1}^{m} \sum\nolimits_{\mu'=1}^{m'} \left( \sum\nolimits_{\nu=1}^{R} \alpha_{\nu,\mu} \alpha'_{\nu,\mu'} \right) \cdot \phi_\mu(\mathbf{x}_{i_1}, \ldots, \mathbf{x}_{i_{|I|}}) \phi'_{\mu'}(\mathbf{x}_{j_1}, \ldots, \mathbf{x}_{j_{|J|}})$$

This is an orthonormal separable decomposition of $h$ (eq. 22), with coefficient matrix $A = \sum_{\nu=1}^{R} \boldsymbol{\alpha}_\nu (\boldsymbol{\alpha}'_\nu)^\top$, where $\boldsymbol{\alpha}_\nu := [\alpha_{\nu,1} \ldots \alpha_{\nu,m}]^\top$ and $\boldsymbol{\alpha}'_\nu := [\alpha'_{\nu,1} \ldots \alpha'_{\nu,m'}]^\top$ for every $\nu \in R$. Obviously the rank of $A$ is no greater than $R$, implying:

$$\frac{\sigma_1^2(A)}{\sigma_1^2(A) + \cdots + \sigma_{\min\{m,m'\}}^2(A)} \geq \frac{1}{R}$$

where as in app. B.1, $\sigma_1(A) \geq \cdots \geq \sigma_{\min\{m,m'\}}(A) \geq 0$ stand for the singular values of $A$. Introducing this inequality into eq. 23 gives:

$$D(h; I, J) = \sqrt{1 - \frac{\sigma_1^2(A)}{\sigma_1^2(A) + \cdots + \sigma_{\min\{m,m'\}}^2(A)}} \leq \sqrt{1 - \frac{1}{R}}$$

The latter holds for any $R \in \mathbb{N}$ that admits eq. 26, so in particular we may take it to be minimal, *i.e.* to be equal to $sep(h; I, J)$ [12], bringing forth the sought after upper bound:

$$D(h; I, J) \leq \sqrt{1 - \frac{1}{sep(h; I, J)}} \qquad (27)$$

By eq. 27, low separation rank implies proximity (in normalized $L^2$ sense) to a separable function. We may use the inequality to translate the upper bounds on separation ranks established for deep and shallow convolutional arithmetic circuits (sec. 5.2 and 5.3 respectively), into upper bounds on normalized $L^2$ distances from separable functions. To completely frame our analysis in terms of the latter measure, a translation of the lower bound on separation ranks of deep convolutional arithmetic circuits (sec. 5.2) is also required. Eq. 27 does not facilitate such translation, and in fact, it is easy to construct functions $h$ whose separation ranks are high yet are very close (in normalized $L^2$ sense) to separable functions. [13] However, as we show in app. B.3 below, the specific lower bound of interest can indeed be translated, and our analysis may entirely be framed in terms of normalized $L^2$ distance from separable functions.

## B.3 LOWER BOUND FOR DEEP CONVOLUTIONAL ARITHMETIC CIRCUITS

Let $h_y \in L^2((\mathbb{R}^s)^N)$ be a function realized by a deep convolutional arithmetic circuit (fig. 1(a) with size-4 pooling windows and $L = \log_4 N$ hidden layers), *i.e.* $h_y$ is given by eq. 2, where $f_{\theta_1} \ldots f_{\theta_M} \in L^2(\mathbb{R}^s)$ are linearly independent representation functions, and $\mathcal{A}^y$ is a coefficient tensor of order $N$ and dimension $M$ in each mode, determined by the linear weights of the network ($\{\mathbf{a}^{l,\gamma}\}_{l,\gamma}, \mathbf{a}^{L,y}$) through the hierarchical decomposition in eq. 3. Rearrange eq. 2 by grouping indexes $d_1 \ldots d_N$ in accordance with the partition $(I, J)$:

$$h_y(\mathbf{x}_1, \ldots, \mathbf{x}_N) = \sum\nolimits_{d_{i_1} \ldots d_{i_{|I|}} = 1}^{M} \sum\nolimits_{d_{j_1} \ldots d_{j_{|J|}} = 1}^{M} \mathcal{A}^y_{d_1 \ldots d_N} \cdot \left( \prod\nolimits_{t=1}^{|I|} f_{\theta_{d_{i_t}}}(\mathbf{x}_{i_t}) \right) \left( \prod\nolimits_{t=1}^{|J|} f_{\theta_{d_{j_t}}}(\mathbf{x}_{j_t}) \right) \qquad (28)$$

---

[12] We disregard the trivial case where $sep(h; I, J) = 0$ ($h$ is identically zero).

[13] This will be the case, for example, if $h$ is given by an orthonormal separable decomposition (eq. 22), with coefficient matrix $A$ that has high rank but whose spectral energy is highly concentrated on one singular value.

Let $m = M^{|I|}$, and define the following mapping:

$$\mu : [M]^{|I|} \to [m] \quad , \quad \mu(d_{i_1}, \ldots, d_{i_{|I|}}) = 1 + \sum_{t=1}^{|I|} (d_{i_t} - 1) \cdot M^{|I|-t}$$

$\mu$ is a one-to-one correspondence between the index sets $[M]^{|I|}$ and $[m]$. We slightly abuse notation, and denote by $(d_{i_1}(\mu), \ldots, d_{i_{|I|}}(\mu))$ the tuple in $[M]^{|I|}$ that maps to $\mu \in [m]$. Additionally, we denote the function $\prod_{t=1}^{|I|} f_{\theta_{d_{i_t}(\mu)}}(\mathbf{x}_{i_t})$, which according to fact 2 in app. A.1 belongs to $L^2((R^s)^{|I|})$, by $\phi_\mu(\mathbf{x}_{i_1}, \ldots, \mathbf{x}_{i_{|I|}})$. In the exact same manner, we let $m' = M^{|J|}$, and define the bijective mapping:

$$\mu' : [M]^{|J|} \to [m'] \quad , \quad \mu'(d_{j_1}, \ldots, d_{j_{|J|}}) = 1 + \sum_{t=1}^{|J|} (d_{j_t} - 1) \cdot M^{|J|-t}$$

As before, $(d_{j_1}(\mu'), \ldots, d_{j_{|J|}}(\mu'))$ stands for the tuple in $[M]^{|J|}$ that maps to $\mu' \in [m']$, and the function $\prod_{t=1}^{|J|} f_{\theta_{d_{j_t}(\mu')}}(\mathbf{x}_{j_t}) \in L^2((R^s)^{|J|})$ is denoted by $\phi'_{\mu'}(\mathbf{x}_{j_1}, \ldots, \mathbf{x}_{j_{|J|}})$. Now, recall the definition of matricization given in sec. 2, and consider $[\![\mathcal{A}^y]\!]_{I,J}$ – the matricization of the coefficient tensor $\mathcal{A}^y$ w.r.t. $(I, J)$. This is a matrix of size $m$-by-$m'$, holding $\mathcal{A}_{d_1 \ldots d_N}$ in row index $\mu(d_{i_1}, \ldots, d_{i_{|I|}})$ and column index $\mu'(d_{j_1}, \ldots, d_{j_{|J|}})$. Rewriting eq. 28 with the indexes $\mu$ and $\mu'$ instead of $(d_{i_1}, \ldots, d_{i_{|I|}})$ and $(d_{j_1}, \ldots, d_{j_{|J|}})$, we obtain:

$$h_y(\mathbf{x}_1, \ldots, \mathbf{x}_N) = \sum_{\mu=1}^{m} \sum_{\mu'=1}^{m'} ([\![\mathcal{A}^y]\!]_{I,J})_{\mu,\mu'} \cdot \phi_\mu(\mathbf{x}_{i_1}, \ldots, \mathbf{x}_{i_{|I|}}) \phi'_{\mu'}(\mathbf{x}_{j_1}, \ldots, \mathbf{x}_{j_{|J|}}) \quad (29)$$

This equation has the form of eq. 22. However, for it to qualify as an orthonormal separable decomposition, the sets of functions $\{\phi_1, \ldots, \phi_m\} \subset L^2((\mathbb{R}^s)^{|I|})$ and $\{\phi'_1, \ldots, \phi'_{m'}\} \subset L^2((\mathbb{R}^s)^{|J|})$ must be orthonormal. If the latter holds eq. 23 may be applied, giving an expression for $D(h_y; I, J)$ – the normalized $L^2$ distance of $h_y$ from the set of separable functions w.r.t. $(I, J)$, in terms of the singular values of $[\![\mathcal{A}^y]\!]_{I,J}$.

We now direct our attention to the special case where $f_{\theta_1} \ldots f_{\theta_M} \in L^2(\mathbb{R}^s)$ – the network's representation functions, are known to be orthonormal. The general setting, in which only linear independence is known, will be treated thereafter. Orthonormality of representation functions implies that $\phi_1 \ldots \phi_m \in L^2((\mathbb{R}^s)^{|I|})$ are orthonormal as well:

$$\begin{aligned}
\langle \phi_\mu, \phi_{\bar{\mu}} \rangle \underset{(1)}{=}& \int \phi_\mu(\mathbf{x}_{i_1}, \ldots, \mathbf{x}_{i_{|I|}}) \phi_{\bar{\mu}}(\mathbf{x}_{i_1}, \ldots, \mathbf{x}_{i_{|I|}}) d\mathbf{x}_{i_1} \cdots d\mathbf{x}_{i_{|I|}} \\
\underset{(2)}{=}& \int \prod_{t=1}^{|I|} f_{\theta_{d_{i_t}(\mu)}}(\mathbf{x}_{i_t}) \prod_{t=1}^{|I|} f_{\theta_{d_{i_t}(\bar{\mu})}}(\mathbf{x}_{i_t}) d\mathbf{x}_{i_1} \cdots d\mathbf{x}_{i_{|I|}} \\
\underset{(3)}{=}& \prod_{t=1}^{|I|} \int f_{\theta_{d_{i_t}(\mu)}}(\mathbf{x}_{i_t}) f_{\theta_{d_{i_t}(\bar{\mu})}}(\mathbf{x}_{i_t}) d\mathbf{x}_{i_t} \\
\underset{(4)}{=}& \prod_{t=1}^{|I|} \left\langle f_{\theta_{d_{i_t}(\mu)}}, f_{\theta_{d_{i_t}(\bar{\mu})}} \right\rangle \\
\underset{(5)}{=}& \prod_{t=1}^{|I|} \left\{ \begin{array}{ll} 1 & , d_{i_t}(\mu) = d_{i_t}(\bar{\mu}) \\ 0 & , \text{otherwise} \end{array} \right\} \\
\underset{(6)}{=}& \left\{ \begin{array}{ll} 1 & , d_{i_t}(\mu) = d_{i_t}(\bar{\mu}) \; \forall t \in [|I|] \\ 0 & , \text{otherwise} \end{array} \right. \\
\underset{(7)}{=}& \left\{ \begin{array}{ll} 1 & , \mu = \bar{\mu} \\ 0 & , \text{otherwise} \end{array} \right.
\end{aligned}$$

(1) and (4) here follow from the definition of inner product in $L^2$ space, (2) replaces $\phi_\mu$ and $\phi_{\bar{\mu}}$ by their definitions, (3) makes use of Fubini's theorem (see Jones (2001)), (5) relies on the (temporary) assumption that representation functions are orthonormal, (6) is a trivial step, and (7) owes to the fact that $\mu \mapsto (d_{i_1}(\mu), \ldots, d_{i_{|I|}}(\mu))$ is an injective mapping. A similar sequence of steps (applied to $\langle \phi'_{\mu'}, \phi'_{\bar{\mu}'} \rangle$) shows that in addition to $\phi_1 \ldots \phi_m$, the functions $\phi'_1 \ldots \phi'_{m'} \in L^2((\mathbb{R}^s)^{|J|})$ will also be orthonormal if $f_{\theta_1} \ldots f_{\theta_M}$ are. We conclude that if representation functions are orthonormal, eq. 29 indeed provides an orthonormal separable decomposition of $h_y$, and the formula in eq. 23 may be applied:

$$D(h_y; I, J) = \sqrt{1 - \frac{\sigma_1^2([\![\mathcal{A}^y]\!]_{I,J})}{\sigma_1^2([\![\mathcal{A}^y]\!]_{I,J}) + \cdots + \sigma_{\min\{m,m'\}}^2([\![\mathcal{A}^y]\!]_{I,J})}} \quad (30)$$

where $\sigma_1([\![\mathcal{A}^y]\!]_{I,J}) \geq \cdots \geq \sigma_{\min\{m,m'\}}([\![\mathcal{A}^y]\!]_{I,J}) \geq 0$ are the singular values of the coefficient tensor matricization $[\![\mathcal{A}^y]\!]_{I,J}$.

In sec. 5.2 we showed that the maximal separation rank realizable by a deep network is greater than or equal to $\min\{r_0, M\}^S$, where $M, r_0$ are the number of channels in the representation and first hidden layers (respectively), and $S$ stands for the number of index quadruplets (sets of the form $\{4k\text{-}3, 4k\text{-}2, 4k\text{-}1, 4k\}$ for some $k \in$

$[N/4]$) that are split by the partition $(I, J)$. To prove this lower bound, we presented in app. A.3 a specific setting for the linear weights of the network $(\{\mathbf{a}^{l,\gamma}\}_{l,\gamma}, \mathbf{a}^{L,y})$ under which $rank[\![\mathcal{A}^y]\!]_{I,J} = \min\{r_0, M\}^S$. Careful examination of the proof shows that with this particular weight setting, not only is the rank of $[\![\mathcal{A}^y]\!]_{I,J}$ equal to $\min\{r_0, M\}^S$, but also, all of its non-zero singular values are equal to one another. [14] This implies that $\sigma_1^2([\![\mathcal{A}^y]\!]_{I,J})/(\sigma_1^2([\![\mathcal{A}^y]\!]_{I,J}) + \cdots + \sigma_{\min\{m,m'\}}^2([\![\mathcal{A}^y]\!]_{I,J})) = \min\{r_0, M\}^{-S}$, and since we currently assume that $f_{\theta_1} \ldots f_{\theta_M}$ are orthonormal, eq. 30 applies and we obtain $D(h_y; I, J) = \sqrt{1 - \min\{r_0, M\}^{-S}}$. Maximizing over all possible weight settings, we arrive at the following lower bound for the normalized $L^2$ distance from separable functions brought forth by a deep convolutional arithmetic circuit:

$$\sup_{\{\mathbf{a}^{l,\gamma}\}_{l,\gamma}, \; \mathbf{a}^{L,y}} D\left(h_y|_{\{\mathbf{a}^{l,\gamma}\}_{l,\gamma}, \mathbf{a}^{L,y}}; I, J\right) \geq \sqrt{1 - \frac{1}{\min\{r_0, M\}^S}} \tag{31}$$

Turning to the general case, we omit the assumption that representation functions $f_{\theta_1} \ldots f_{\theta_M} \in L^2(\mathbb{R}^s)$ are orthonormal, and merely rely on their linear independence. The latter implies that the dimension of $span\{f_{\theta_1} \ldots f_{\theta_M}\}$ is $M$, thus there exist orthonormal functions $\varphi_1 \ldots \varphi_M \in L^2(\mathbb{R}^s)$ that span it. Let $F \in \mathbb{R}^{M \times M}$ be a transition matrix between the bases – the matrix defined by $\varphi_c = \sum_{d=1}^{M} F_{c,d} \cdot f_{\theta_d}, \forall c \in [M]$. Suppose now that we replace the original representation functions $f_{\theta_1} \ldots f_{\theta_M}$ by the orthonormal ones $\varphi_1 \ldots \varphi_M$. Using the latter, the lower bound in eq. 31 applies, and there exists a setting for the linear weights of the network – $\{\mathbf{a}^{l,\gamma}\}_{l,\gamma}, \mathbf{a}^{L,y}$, such that $D(h_y; I, J) \geq \sqrt{1 - \min\{r_0, M\}^{-S}}$. Recalling the structure of convolutional arithmetic circuits (fig. 1(a)), one readily sees that if we return to the original representation functions $f_{\theta_1} \ldots f_{\theta_M}$, while multiplying conv weights in hidden layer 0 by $F^\top$ (i.e. mapping $\mathbf{a}^{0,\gamma} \mapsto F^\top \mathbf{a}^{0,\gamma}$), the overall function $h_y$ remains unchanged, and in particular $D(h_y; I, J) \geq \sqrt{1 - \min\{r_0, M\}^{-S}}$ still holds. We conclude that the lower bound in eq. 31 applies, even if representation functions are not orthonormal.

To summarize, we translated the lower bound from sec. 5.2 on the maximal separation rank realized by a deep convolutional arithmetic circuit, into a lower bound on the maximal normalized $L^2$ distance from separable functions (eq. 31). This, along with the translation of upper bounds facilitated in app. B.2, implies that the analysis carried out in the paper, which studies correlations modeled by convolutional networks through the notion of separation rank, may equivalently be framed in terms of normalized $L^2$ distance from separable functions. We note however that there is one particular aspect in our original analysis that does not carry through the translation. Namely, in sec. 5.1 it was shown that separation ranks realized by convolutional arithmetic circuits are maximal almost always, *i.e.* for all linear weight settings but a set of (Lebesgue) measure zero. Put differently, for a given partition $(I, J)$, the maximal separation rank brought forth by a network characterizes almost all functions realized by it. An equivalent statement does not hold with the continuous measure of normalized $L^2$ distance from separable functions. The behavior of this measure across the hypotheses space of a network is non-trivial, and forms a subject for future research.

## C IMPLEMENTATION DETAILS

In this appendix we provide implementation details omitted from the description of our experiments in sec. 7. Our implementation, available online at `https://github.com/HUJI-Deep/inductive-pooling`, is based on the SimNets branch (Cohen et al. (2016a)) of Caffe toolbox (Jia et al. (2014)). The latter realizes convolutional arithmetic circuits in log-space for numerical stability.

When training convolutional arithmetic circuits, we followed the hyper-parameter choices made by Sharir et al. (2016). In particular, our objective function was the cross-entropy loss with no $L^2$ regularization (*i.e.* with weight decay set to 0), optimized using Adam (Kingma and Ba (2014)) with step-size $\alpha = 0.003$ and moment decay rates $\beta_1 = \beta_2 = 0.9$. 15000 iterations with batch size 64 (48 epochs) were run, with the step-size $\alpha$ decreasing by a factor of 10 after 12000 iterations (38.4 epochs). We did not use dropout (Srivastava et al. (2014)), as the limiting factor in terms of accuracies was the difficulty of fitting training data (as opposed to overfitting) – see fig. 3.

For training the conventional convolutional rectifier networks, we merely switched the hyper-parameters of Adam to the recommended settings specified in Kingma and Ba (2014) ($\alpha = 0.001, \beta_1 = 0.9, \beta_2 = 0.999$), and set weight decay to the standard value of 0.0001.

---

[14] To see this, note that with the specified weight setting, for every $n \in [N/4]$, $[\![\phi^{1,1}]\!]_{I_{1,n}, J_{1,n}}$ has one of two forms: it is either a non-zero (row/column) vector, or it is a matrix holding 1 in several entries and 0 in all the rest, where any two entries holding 1 reside in different rows and different columns. The first of the two forms admits a single non-zero singular value. The second brings forth several singular values equal to 1, possibly accompanied by null singular values. In both cases, all non-zero singular values of $[\![\phi^{1,1}]\!]_{I_{1,n}, J_{1,n}}$ are equal to one another. Now, since $[\![\mathcal{A}^y]\!]_{I,J} = \odot_{n=1}^{N/4} [\![\phi^{1,1}]\!]_{I_{1,n}, J_{1,n}}$, and since the Kronecker product multiplies singular values (see Bellman (1970)), we have that all non-zero singular values of $[\![\mathcal{A}^y]\!]_{I,J}$ are equal, as required.

## D    MORPHOLOGICAL CLOSURE

The synthetic dataset used in our experiments (sec. 7) consists of binary images displaying different shapes (blobs). One of the tasks facilitated by this dataset is the detection of morphologically closed blobs, *i.e.* of images that are relatively similar to their morphological closure. The procedure we followed for computing the morphological closure of a binary image is:

1. Pad the given image with background (0 value) pixels

2. Morphological dilation: simultaneously turn on (set to 1) all pixels that have a (left, right, top or bottom) neighbor originally active (holding 1)

3. Morphological erosion: simultaneously turn off (set to 0) all pixels that have a (left, right, top or bottom) neighbor currently inactive (holding 0)

4. Remove pixels introduced in padding

It is not difficult to see that any pixel active in the original image is necessarily active in its closure. Moreover, pixels that are originally inactive yet are surrounded by active ones will also be turned on in the closure, hence the effect of "gap filling". Finally, we note that the particular sequence of steps described above represents the most basic form of morphological closure. The interested reader is referred to Haralick et al. (1987) for a much more comprehensive introduction.

