# Peer review of "Inductive Bias of Deep Convolutional Networks through Pooling Geometry"

_ICLR 2017 — accepted_

[Official Review · AnonReviewer2 · rating 6 · confidence 3 · 16 Dec 2016]
**Interesting analysis**

This paper addresses the question of which functions are well suited to deep networks, as opposed to shallow networks.  The basic intuition is convincing and fairly straightforward.  Pooling operations bring together information.  When information is correlated, it can be more efficiently used if the geometry of pooling regions matches the correlations so that it can be brought together more efficiently.  Shallow networks without layers of localized pooling lack this mechanism to combine correlated information efficiently.

The theoretical results are focused on convolutional arithmetic circuits, building on prior theoretical results of the authors.  The results make use of the interesting technical notion of separability, which in some sense measures the degree to which a function can be represented as the composition of independent functions.  Because separability is measured relative to a partition of the input, it is an appropriate mechanism for measuring the complexity of functions relative to a particular geometry of pooling operations.  Many of the technical notions are pretty intuitive, although the tensor analysis is pretty terse and not easy to follow without knowledge of the authors’ prior work.

In some sense the comparison between deep and shallow networks is somewhat misleading, since the shallow networks lack a hierarchical pooling structure.  For example, a shallow convolutional network with RELU and max pooling does not really make sense, since the max occurs over the whole image.  So it seems that the paper is really more of an analysis of the effect of pooling vs. not having pooling.  For example, it is not clear that a deep CNN without pooling would be any more efficient than a shallow network, from this work.

It is not clear how much the theoretical results depend on the use of a model with product pooling, and how they might be extended to the more common max pooling.  Even if theoretical results are difficult to derive in this case, simple illustrative examples might be helpful.  In fact, if the authors prepare a longer version of the paper for a journal I think the results could be made more intuitive if they could add a simple toy example of a function that can be efficiently represented with a convolutional arithmetic circuit when the pooling structure fits the correlations, and perhaps showing also how this could be represented with a convolutional network with RELU and max pooling.

I would also appreciate a more explicit discussion of how the depth of a deep network affects the separability of functions that can be represented.  A shallow network doesn’t have local pooling, so the difference between deep and shallow if perhaps mostly one of pooling vs. not pooling.  However, practitioners find that very deep networks seem to be more effective than “deep” networks with only a few convolutional layers and pooling.  The paper does not explicitly discuss whether their results provide insight into this behavior.

Overall, I think that the paper attacks an important problem in an interesting way.  It is not so convincing that this really gets to the heart of why depth is so important, because of the theoretical limitation to arithmetic circuits, and because the comparison is to shallow networks that are without localized pooling.

[Official Review · AnonReviewer3 · rating 7 · confidence 3 · 20 Dec 2016]
**Huge algebraic machinery, so far only used to perform intuitive model selection, but promising direction.**

The paper provides a highly complex algebraic machinery to analyze the type of functions covered by convolutional network. As in most attempts  in this direction in the literature, the ideal networks described in paper, which have to be interpretable as polynomials over tensors, do not match the type of CNNs used in practice: for instance the Relu non-linearity is replaced with a product of linear functions (or a sum of logs).

While the paper is very technical to read, every concept is clearly stated and mathematical terminology properly introduced. Still, I think some the authors could make some effort to make the key concepts more accessible, and give a more intuitive understanding of what the separation rank means rather before piling up different mathematical interpretation.
My SVM-era algebra is quite rusted, and I am not familiar with the separation rank framework: it would have been much easier for me to first fully understand a simple and gentle case (shallow network in section 5.3), than the general deep case.

To summarize my understanding of the key theorem 1 result:
- The upper bound of the separation rank is used to show that in the shallow case, this rank grows AT MOST linearly with the network size (as measured by the only hidden layer). So exponential network sizes are caused by this rank needing to grow exponentially, as required by the partition.
- In the deep case, one also uses the case that the upper bound is linear in the size of the network (as measured by the last hidden layer), however, this situation is caused by the selection of a partition (I^low, J^high), and the maximal rank induced by this partition is only linear anyway, hence the network size can remain linear.

If tried my best to summarize the key point of this paper and still probably failed at it, which shows how complex is this notion of partition rank, and that its linear growth with network size can either be a good or bad thing depending on the setting. Hopefully, someone will come one day with an explanation that holds in a single slide.



While this is worth publishing as conference paper in its present form, I have two suggestions that, IMHO, would make this work more significant:

On the theory side, we are still very far from the completeness of the PAC bound papers of the "shallow era". In particular, the non-probabilistic lower and upper bound in theorem 1 are probably loose, and there is no PAC-like theory to tell us which one to use and what is the predicted impact on performance (not just the intuition). Also, in the prediction of the inductive bias, the other half is missing. This paper attempts to predict the maximal representation capacity of a DNN under bounded network size constraints, but one of the reason why this size has to be bounded is overfitting (justified by PAC or VC-dim like bounds). If we consider the expected risk as  the sum of the empirical risk and the structural risk, this paper only seems to address fully the empirical risk minimization part, freezing the structural risk. 

On the practice side, an issue is that experiments in this paper mostly confirm what is obvious through intuition, or some simpler form of reasonings. For instance to use convolutions that join pixels which are symmetrical in images to detect symmetry. Basic hand-crafted pattern detectors, as they have been used in computer vision for decades, would just do the job. What would be a great motivation for using this framework is if it answered questions that simple human intuition cannot, and for which we are still in the dark: one example I could think of in the recent use of gated convolutions 'a trous' for 1D speech signal, popularized in Google WaveNet (

[Official Review · AnonReviewer4 · rating 7 · confidence 5 · 09 Jan 2017]
**Promising approach to show why deep CNN works well in practice**

This paper investigates the fact why deep networks perform well in practice and how modifying the geometry of pooling can make the polynomially
sized deep network to provide a function with exponentially high separation rank (for certain partitioning.)

In the authors' previous works, they showed the superiority of deep networks over shallows when the activation function is ReLu and the pooling is max/mean pooling but in the current paper there is no activation function after conv and the pooling is just a multiplication of the node values. Although for the experimental results they've considered both scenarios. 

Actually, the general reasoning for this problem is hard, therefore, this drawback is not significant and the current contribution adds a reasonable amount of knowledge to the literature. 

This paper studies the convolutional arithmetic circuits and shows how this model can address the inductive biases and how pooling can adjust these biases. 

This interesting contribution gives an intuition about how deep network can capture the correlation between the input variables when its size is polynomial but and correlation is exponential.

It worth to note that although the authors tried to express their notation and definitions carefully where they were very successful, it would be helpful if they elaborate a bit more on their definitions, expressions, and conclusions in the sense to make them more accessible.

[Final Decision · Program Chairs · 06 Feb 2017]
**ICLR committee final decision**

The paper uses the notion of separation rank from tensor algebra to analyze the correlations induced through convolution and pooling operations. They show that deep networks have exponentially larger separation ranks compared to shallow ones, and thus, can induce a much richer correlation structure compared to shallow networks. It is argued that this rich inductive bias is crucial for empirical success.
 
 The paper is technically solid. The reviewers note this, and also make a few suggestions on how to make the paper more accessible. The authors have taken this into account. In order to bridge the gap between theory and practice, it is essential for theory papers to be accessible.
 
 The paper covers related work pretty well. One aspect is misses is the recent geometric analysis of deep learning. Can the algebraic analysis be connected to geometric analysis of deep learning, e.g. in the following paper?